# The application and modification of WRF-Hydro/Glacier to a cold-based Antarctic glacier

Tamara Pletzer[1], Jonathan P. Conway[2], Nicolas J. Cullen[1], Trude Eidhammer[3], and Marwan Katurji[4]

[1]School of Geography, University of Otago, Ōtepoti/Dunedin, New Zealand
[2]National Institute of Water and Atmospheric Research, Lauder, New Zealand
[3]National Center for Atmospheric Research, P.O. Box 3000, Boulder, CO 80307, USA
[4]School of Earth and Environment, University of Canterbury, Ōtautahi/Christchurch, New Zealand

**Correspondence:** Tamara Pletzer (tamara.pletzer@postgrad.otago.ac.nz)

**Abstract.** The McMurdo Dry Valleys (MDV) are home to a unique microbial ecosystem dependent on the availability of freshwater. It is a polar desert and freshwater originates almost entirely from surface and near-surface melt of the cold-based glaciers. Understanding the future evolution of these environments requires the simulation of the full chain of physical processes from net radiative forcing, surface energy balance, melt, runoff and the transport of meltwater in stream channels from the glaciers to the terminal lakes where the microbial community resides. To establish a new framework to do this, we present the first application of WRF-Hydro/Glacier in the MDV, which as a fully distributed hydrological model, has the capability to resolve the pathways of meltwater from the glaciers to the bare-land surfaces that surround them. Given meltwater generation in the MDV is almost entirely dependent on small changes in the energy balance of the glaciers, the aim of this study is to optimise the multi-layer snowpack scheme that is embedded into WRF-Hydro/Glacier to ensure the feedbacks between albedo, snowfall and melt are fully resolved. To achieve this, WRF-Hydro/Glacier is implemented at a point scale using automatic weather station data on Commonwealth Glacier to physically model the onset, duration and end of melt over a 7-month period (1 August 2021 to 28 February 2022). To resolve the limited energetics controlling melt, it was necessary to (1) limit the percolation of meltwater through the ice layers in the multi-layer snowpack scheme and (2) optimise the parameters controlling the albedo of both snow and ice over the melt season based on observed spectral signatures of albedo. These modifications enabled the variability of broadband albedo over the melt season to be accurately simulated, as well as ensuring modelled surface and near-surface temperatures, surface height change and runoff were fully resolved. By establishing a new framework that couples a detailed snowpack model to a fully distributed hydrological model, this work provides a stepping stone to model the spatial and temporal variability of melt and streamflow in the future, which will enable some of the unknown questions about the hydrological connectivity of the MDV to be answered.

## 1 Introduction

Terrestrial Antarctic ecosystems exist almost entirely in ice free regions. Under the highest emissions scenario, RCP 8.5, climate studies show that these ice-free regions may increase from 1% of Antarctica to almost 25% by the end of the century (Lee et al., 2017). Given the anticipated increase in ice-free regions in Antarctica, there is an urgent need to better understand the

sensitivity of the McMurdo Dry Valleys (MDV) to climate variability and change. The MDV are currently the largest ice-free area in Antarctica and home to a unique microbial ecosystem that resides in a system of streams and lakes situated in the valley floors. This ecosystem is dependent on fresh water that is sourced from glacial melt (Gooseff et al., 2011) for survival. It is expected that the biogeography of this ecosystem will be altered in response to the larger changes in climate impacting Antarctica, thus it is key to understand and simulate how atmospheric warming will impact glacier melt and the hydrological connectivity of this unique environment.

Unlike most mid-latitude glaciers, the glaciers in the MDV are mainly cold-based, meaning that the subsurface temperatures are below the pressure melting point and ice is frozen to the bedrock (Fitzsimons, 1996; Gooseff et al., 2011; Fountain et al., 2016). MacDonell (2008) was the first to study the full hydrological drainage system and developed a conceptual modelling framework for cold-based glacial meltwater drainage processes on Lower Wright Glacier in the Wright Valley. This study identified the need for a physically based, rather than empirical, model to simulate the complex drainage systems of MDV glaciers (MacDonell, 2008). Hoffman (2011) was the first to implement a distributed surface energy balance model and investigated the spatial and temporal distribution of mass and energy exchanges of several glaciers in Taylor Valley. This study showed that penetrating shortwave radiation and meltwater drainage are important for accurately modelling surface ablation on glaciers in the MDV (Hoffman, 2011). Cross et al. (2022) extended this work by coupling the glacier energy balance model used by Hoffman (2011) with a lake energy balance model to assess lake sublimation and the water budget. Whilst both Hoffman (2011) and Cross et al. (2022) model glacial runoff, neither explicitly model the off-glacier processes in the hydrological system, such as streams or soil moisture.

Similar to mid-latitude glaciers, glacial melt in the MDV is driven primarily by net radiation, which is sensitive to variability in solar radiation. What is different in the MDV is that energy and mass exchanges are often very small, with minor changes in the surface energetics capable of shifting the hydrological system from a frozen state to one that is melting. Thus, small changes in energy (for example, through albedo) can have a very large impact on melt generation (Hoffman et al., 2008; Macdonell et al., 2013). It has also been observed that solar radiation penetrates the top 5-15 cm of the snow or ice surfaces on the glaciers in the MDV (Hoffman et al., 2014) and this near-surface layer can retain heat longer than the surface due to the solid-state greenhouse effect (Brandt and Warren, 1993). This effect extends the duration of melt events and model simulations suggest melt from this layer can be an order of magnitude larger than surface melt (Hoffman et al., 2014). Despite solar radiation being crucial for modelling melt, no study has attempted to physically simulate the variability of albedo over the duration of a full melt season in the MDV. To ensure the magnitude and duration of melt is captured sufficiently in a low energy, polar environment like the MDV, it is critical that glacier albedo is modelled accurately.

To resolve the hydrological connectivity in the MDV it is necessary to identify the pathways of meltwater from the glaciers to the bare-land surfaces that surround them, including understanding how water is channelled into stream networks and stored in the numerous closed-basin lakes. The WRF-Hydro/Glacier modelling framework (Gochis et al., 2020; Eidhammer et al.,

2021) provides an opportunity to physically model the hydrological cycle in the MDV due to its ability to resolve the connections between the atmosphere, glaciers, bare-land surfaces and soils, stream channels and lakes (the hydrological reservoirs). WRF-Hydro/Glacier contains a detailed snowpack model (Crocus), which is embedded into a distributed hydrological model (WRF-Hydro). Importantly, the model provides enough flexibility to be able to resolve the meltwater pathways from the glaciers to the surrounding landscape, which is critical given the glaciers are the primary hydrologic reservoirs and controlled by the daily, seasonal, and annual cycles of the surface energy balance (Gooseff et al., 2011). Importantly, WRF-Hydro/Glacier can be applied at small catchment scales up to continental scale domains, providing fully-distributed hydrological modelling opportunities in the MDV and the larger surrounding regions. Eidhammer et al. (2021) implemented WRF-Hydro/Glacier on a temperate Norwegian glacier and demonstrated that the model was capable of simulating the mass balance, snow depth, surface albedo and runoff when compared to observations. However, the environmental setting of the cold-based MDV glaciers is vastly different from the temperate glaciers in Norway, with the dry polar climate of the MDV ensuring that summer melt generated from variability in the surface energy balance has a much stronger control on the hydrological cycle than precipitation (Gooseff et al., 2011).

Given the importance of the surface energy balance to meltwater generation, the modelling of albedo and near-surface melt on the MDV glaciers is critical. The detailed snowpack scheme in Crocus, which is embedded in WRF-Hydro/Glacier, has been implemented in a range of environments (Vionnet et al., 2012), including extremely cold (well below the melting point) conditions over the Antarctic ice sheet (such as Dome C) (Brun et al., 2011) and high precipitation (both rain and snow) and relatively warm and melting conditions on temperate glaciers (Eidhammer et al., 2021). However, the challenge in implementing it on glaciers in the MDV is that the surface energy balance is more dominant than precipitation in governing variability in mass balance, largely through its influence on controlling melt and the associated feedbacks on surface albedo. The implication of this is that runoff and streamflow are entirely sourced from glacier melt rather than from precipitation (rainfall), as is common on temperate glaciers. In this context, the aim of this study is to optimise a multi-layer snowpack scheme (Crocus), that is embedded in WRF-Hydro/Glacier, to resolve the onset, duration and end of melt over a cold-based glacier in the MDV of Antarctica. If melt is sufficiently resolved in this modelling framework, it will allow the pathways of meltwater to the surrounding landscape (the other hydrological reservoirs) to be resolved at different spatial and temporal scales in future applications of WRF-Hydro/Glacier.

To achieve our primary aim, WRF-Hydro/Glacier is implemented at a point on Commonwealth Glacier in Taylor Valley, forced by automatic weather station data and tested against observations of broadband albedo, surface and near-surface ice temperatures, surface height change and streamflow. Given it is the first time this modelling framework has been implemented in this unique environmental setting (cold-based glacier, energy balance dominates over precipitation, limited opportunities for melting threshold to be reached) it was necessary to modify the percolation of water through the glacier and the spectral albedo scheme to accurately simulate the feedbacks between albedo, snowfall and melt. The limited energetics and complicated pathways for water transport in this environment make testing at a point scale using observational rather than modelled input data

a critical first step towards modelling the full hydrological connectivity of glacial meltwater in the MDV.


The next section describes the WRF-Hydro/Glacier model setup and initialization. Section 3 describes the site and the observational data used to force and validate the model. Section 4 details the modifications to the Crocus snowpack meltwater drainage and spectral albedo schemes necessary to adapt the model to the unique cold-based glacial environment of the MDV. Section 5 compares the performance of the original and modified versions of the model to observational data over the 2021/22

melt season, while Section 6 reflects on the significant advances made to WRF-Hydro/Glacier in this study and the platform it provides to resolve the full hydrological system in the MDV.

## 2 Model description

### 2.1 WRF-Hydro/Glacier

WRF-Hydro is a multi-scale and spatially distributed hydro-meteorological modelling system developed by NCAR (Gochis

et al., 2020). The model links a column land surface model (Noah-MP) with subsurface flow, routing, overland flow, channel routing and water management modules (i.e. reservoir). The model can be either forced by meteorological data or gridded atmospheric data. A schematic of the modelling system can be seen in Figure 1. WRF-Hydro is currently the National Water Model in the contiguous USA. It has applications from flood forecasting to regional hydroclimate impact assessments and has been validated extensively across six continents (e.g. Senatore et al. (2015); Li et al. (2017); Xiang et al. (2017); Kerandi

et al. (2018); Lahmers et al. (2021); Pal et al. (2021); Shafqat Mehboob et al. (2022)). More recently, Eidhammer et al. (2021) developed the WRF-Hydro/Glacier model by embedding the detailed Crocus snowpack model into the Noah-MP land surface model in order to simulate the energy and mass balance over glacial surfaces. The snow module in Noah-MP does not allow the glacier to decrease in mass once the snow layers are melted because it represents the glacier as a bare ice land surface category with constant albedo, roughness length and heat conductivity. This representation does not allow melt and thus, was

not sufficient to model glacial melt and runoff (Eidhammer et al., 2021).

Crocus is a detailed column snowpack model that was developed initially for avalanche forecasting in Col de Porte, France by the Centre d'Etudes de la Neige of Météo-France (Brun et al., 1989, 1992). The model version implemented in WRF-Hydro/Glacier is described in Vionnet et al. (2012). This version was chosen by Eidhammer et al. (2021) due to extensive

validation and its suitability for embedding into the Noah-MP land surface scheme. The model physics include schemes for snow metamorphism, compaction, albedo, penetrating solar radiation, energy balance fluxes, heat diffusion, melt, refreezing, percolation, runoff and sublimation/deposition. Crocus simulates state variables in layers: heat content, thickness, density, age, history of snow and two snow grain properties that describe the dendricity, sphericity and grain size of the snow or ice crystals over a prescribed number of layers. All other variables such as snow temperature and liquid water content are calculated from

the state variables. The number of and thickness of vertical layers in Crocus changes dynamically with time. Users define a maximum number of layers ($n \geq 3$) and when snowfall occurs, a new layer is added with a set of fresh snow characteristics.

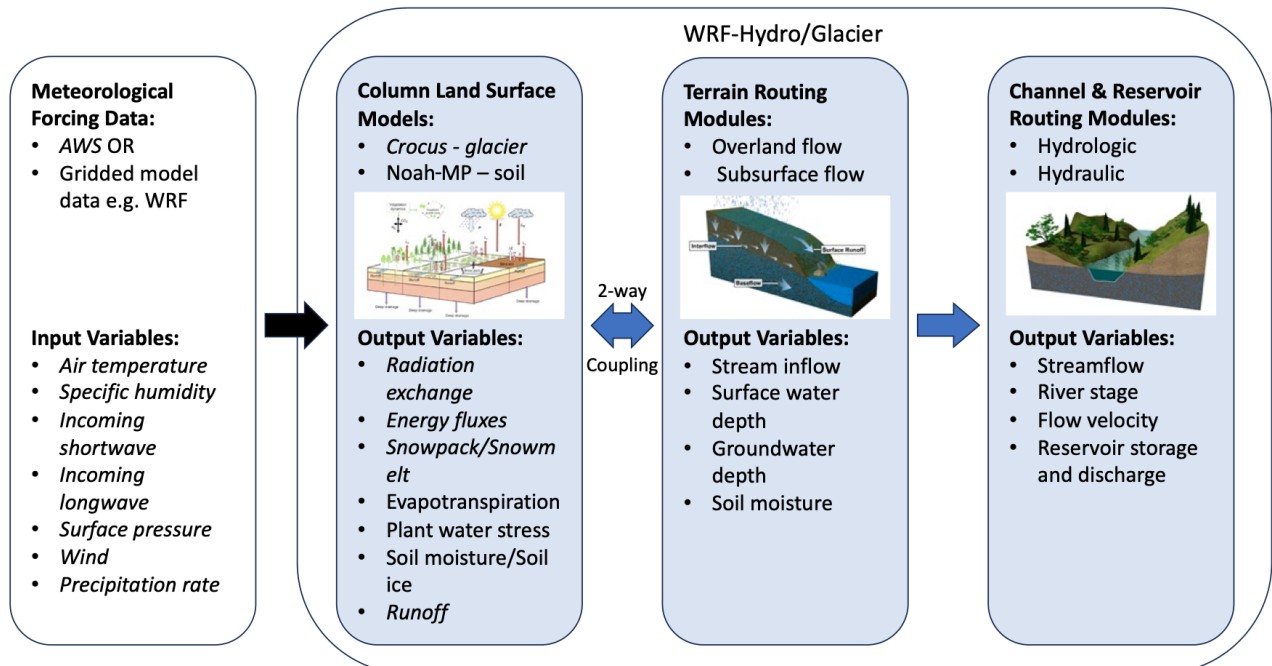

**Figure 1.** Schematic of the WRF-Hydro/Glacier modelling framework. Modules and variables used in this study are displayed in *italics*. Adapted from NCAR (https://ral.ucar.edu/sites/default/files/public/WRFHydroPhysicsComponentsandOutputVariables.png, last access: 8 August 2023).

Over time, layers may merge with the layer below if the snow grain properties become the same. The layers at the top of the snowpack tend to be thinner to better solve the surface energy balance equation. Further information can be found in Vionnet et al. (2012) and Eidhammer et al. (2021).


## 3   Methods and data

### 3.1   Site description

Commonwealth Glacier is located on the eastern side of Taylor Valley approximately 4 km from McMurdo Sound (Fig. 2). It is a piedmont glacier, terminating in cliffs and is characterized by a smooth surface. The mean annual temperature from 135   1986–2017 is -17.6 °C, however in the summer temperatures can reach a maximum of 7.8 °C and in winter the temperature can descend to a minimum of -45.0 °C (Obryk et al., 2020). The wind directions are characterized by a bimodal distribution in the summer, mainly a daytime easterly sea breeze coming from McMurdo Sound and northerly down-glacier wind due to

**Table 1.** Variables and instruments on CWG AWS installed on 1 December 2021. The instruments sample every 1 minute and averages are taken every 30 minutes. Data are stored on a CR1000 data logger. Radiation accuracies for shortwave and longwave radiation are given in terms of the estimated accuracy of daily totals (EADT).

| Variable | Instrument | Accuracy | Sensor height (m) |
|---|---|---|---|
| Wind speed | Young Wind Anemometer 05108-L40 | $\pm 0.3$ m s$^{-1}$ or 1 % | 2.3 |
| Wind direction | Young Wind Anemometer 05108-L40 | $\pm 3.0$ ° | 2.3 |
| Air temperature | Vaisala HMP 155 | $\pm 0.17$ °C at 20 °C | 1.4 |
| Relative humidity | Vaisala HMP 155 | $\pm 1$ (%RH 0-90) at 20 °C | 1.4 |
| | | $\pm 1.7$ (%RH 90-100) at 20 °C | |
| Ice temperature | Type T Thermocouples | $\pm 1$ °C or 0.75% | -0.05, -0.1, -0.2, -0.5, -1.0 |
| | | | and -2.0 (initial heights) |
| Shortwave radiation (incoming and outgoing) | Apogee Pyranometer SN500SS | $< 5$ % EADT | 1.8 |
| Longwave radiation (incoming and outgoing) | Apogee Pyranometer SN500SS | $\pm 5$ % EADT | 1.8 |
| Distance to surface | SR50 Sonic ranger | $\pm 1$ cm | 1.7 |
| | | or 0.4 % of distance to target | |
| Pressure | Vaisala PTB110 | $\pm 0.3$ hPa at 20 °C | 0.5 |

the low sun angle and the temperature differences between the glacier and bare land and soils on the valley floor at night time (Fountain and Doran, 2004). In the winter, this down-glacier wind becomes more persistent as the ice cools faster than the bare land and soil. There is also a low frequency westerly in both summer and winter that is a down-valley wind (Nylen et al., 2004). The down valley winds can either be a föhn wind generated by a synoptic cyclone or a katabatic descending from the Antarctic Plateau (Speirs et al., 2010; Steinhoff et al., 2013). Wind speeds on average are 2.2 m s$^{-1}$, with the strongest reported reaching a maximum of 44.5 m s$^{-1}$ (Obryk et al., 2020).

### 3.2 Automatic weather stations

Data from two automatic weather stations (AWSs) are used in this study. The CWG AWS was installed on 1 December 2021 and is used for tuning and validation. It is located at $-77.56485°$, $163.2776°$ at 280 meters in elevation. CWG AWS measures air temperature, relative humidity, wind speed and direction, air pressure, near-surface ice temperatures and the incoming and outgoing components of shortwave and longwave radiation (Fig. 2c). Table 1 shows the instruments and accuracy of each sensor. The instruments are sampled every 1 minute and averages are taken every 30 minutes. Data are stored on a CR1000 data logger.

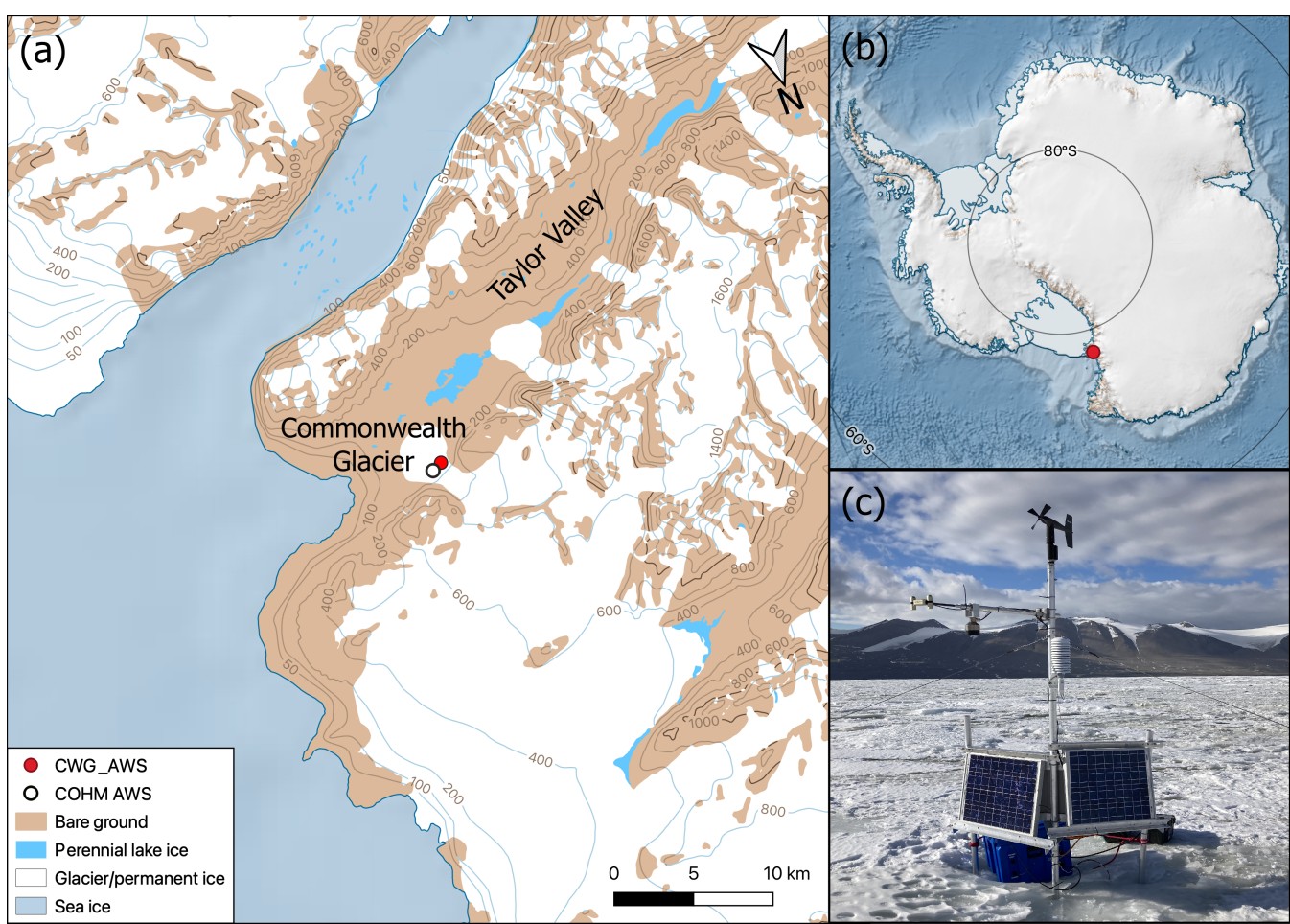

**Figure 2.** (a) Map of Commonwealth Glacier with respect to Taylor Valley with CWG AWS in red filled circle. COHM AWS is indicated by a black open circle located 140 meters to the east of CWG AWS. (b) Taylor Valley with respect to the continent. (c) The CWG AWS on 31 January 2022. Detailed basemaps from RAMP2 Hillshade, ETOPO1/IBCSCO/RAMP2 Hillshade and Elevation model. This map was made using the Quantarctica QGIS package collated by the Norwegian Polar Institute (https://www.npolar.no/quantarctica/, last access: 9 April 2023).

The MDV Long Term Ecological Research Project station on Commonwealth Glacier (COHM AWS) is located at $-77.563712°$, $163.280145°$ and 290 m in elevation (Doran and Fountain, 2022). COHM AWS is located 140 meters east of CWG AWS. The accuracy of the sensors are similar to CWG AWS and the instruments are detailed in Gooseff et al. (2022).

155

**Table 2.** Summary of time periods and forcing data used for model experiment.

|  | Time period | Forcing data |
| --- | --- | --- |
| Spin up | 1 August - 30 November 2021 | COHM AWS |
| Albedo tuning | 1 - 31 December 2021 | CWG AWS |
| Testing | 1 January - 28 February 2022 | CWG AWS |

## 3.3 Model forcing

The model is forced at an hourly time step by observational data from the COHM AWS and CWG AWS. For the model spin-up, the MDV Long Term Ecological Research Project station on Commonwealth Glacier (COHM AWS) (Doran and Fountain, 2022) is used because it is before the CWG AWS was installed. The time periods and forcing data used are shown in Table 2.

The following meteorological variables were used: incoming shortwave radiation (W m$^{-2}$), incoming longwave radiation (W m$^{-2}$), air temperature at 2 meters (K), the meridional and zonal wind vector components (m s$^{-1}$), specific humidity (kg kg$^{-1}$), surface pressure (hPa) and precipitation (mm water-equivalent s$^{-1}$). The data from the AWSs were processed similarly to Gillett and Cullen (2011), where specific humidity was calculated with respect to water and ice depending on whether the air temperature was above or below the melting point. Pressure for the spin-up period was obtained from the nearest grid cell to the CWG AWS in Antarctic Mesoscale WRF Prediction System (AMPS) atmospheric data as there was no barometer on the COHM AWS.

Given that there was no precipitation sensor on either AWS, precipitation was obtained using changes in snow height measurements from the SR50 sensor on COHM and CWG AWSs. Fountain et al. (2010) calculated precipitation in the MDV by taking the daily average change in snow height. If the change in snow height between days was greater than 5 mm, this was taken as precipitation. We attempted this method by Fountain et al. (2010), but found that it filtered out all of the summer snowfall events (some of which were witnessed during field work). Instead, we opted to fit a K nearest neighbors regression (n=30 and uniform weights) to the 30 minute snow height timeseries in order to filter the noise from the sensor. This method is similar to a rolling mean and smooths the snow height timeseries. Values below 0.3 mm (0.4 mm for January and February) of change in snow height during a 30 minute period were removed to filter noise from the sensor. This threshold was tuned to observations of snow events from the 2021/22 field campaign and observed albedo. Next, the positive increases in the hourly sum of surface height change were converted to mm water-equivalent using a fresh snow density of 150 kg m$^{-3}$ to convert height change to the mass of snow.

The forcing data are displayed in Fig. 3. Meteorological conditions vary seasonally with a stark difference in incoming short-wave radiation between the polar night and summer. This controls the surface temperatures that vary between the melting point

in the summer and a minimum of -43 °C in August, which is often the coldest month in the MDV. Relative humidities are low and snowfall amounts are very low, with a higher frequency near the middle of the summer. Wind direction varies between northeasterlies (sea breeze), northerlies (down-glacier katabatic) and westerlies (föhn), with the greatest wind speeds occurring when there is a westerly. The average pressure over the period is 950 hPa.

Comparing the 2021/22 season to the 1999-2022 long term average over December and January (Hofsteenge et al., 2023, *in review*), we find that air temperature, wind and albedo were close to average. Air temperatures were 0.2 °C below average, windspeed was 0.1 m s$^{-1}$ below average and albedo was 0.02 below average. There was slightly less cloud cover as incoming shortwave radiation was 43.4 W m$^{-2}$ (13.9%) above average and incoming longwave radiation was 6.9 W m$^{-2}$ (-3.0%) below average.

## 3.4 Implementation and initialization

In this model experiment, we used the Crocus snowpack model embedded in WRF-Hydro/Glacier version 5.2.0 (Eidhammer et al., 2021). Crocus was forced by observed meteorological data at an hourly time step and analyzed at a point on Commonwealth Glacier. The high-quality observational data obtained on Commonwealth Glacier reduces uncertainty that might be introduced by using model or gridded data as meteorological forcing data. Crocus was initialized with 40 layers (as in Eidhammer et al. (2021)) and constant thickness of 50 meters for simplicity and in line with terminal cliff height observations. Snow depth was initialized as 0 m and the layers were initialized with a constant density of 900 kg m$^{-3}$. The glacier temperatures were initialized at -18 °C for each layer and with a constant temperature of -18 °C at the soil-glacier boundary since the temperature at this depth is expected to be the annual mean air temperature (Fountain et al., 1998). More information on the model configuration can be found in the namelists provided in *Code and data availability*.

The WRF-Hydro/Glacier spin-up period was 1 August to 30 November 2021 to obtain a realistic ice temperature profile at the start of the simulation. We analyzed ice temperatures at the end of the spin-up period and found that the difference between observed and modelled ice temperatures at a depth of 0.05, 0.1, 0.2, 0.5 and 2.0 meters at the beginning of December were less than 1 °C, which is within the sensor uncertainty shown in Table 1. Given this agreement, we concluded the spin-up time is sufficient for the model testing in this evaluation (see Section 5.2 for further discussion). Following the spin-up period, the different model configurations were validated from 1 December 2021 to 28 February 2022. This period was chosen because there is no observed streamflow measurements outside of these months. Ice surface temperatures reach the melting point for the first time in December and the temperatures switch to a primarily cooling regime in February, descending well below the melting point.

## 3.5 Validation data

WRF-Hydro/Glacier is validated using surface temperature, internal ice temperatures, albedo, surface height change and streamflow over the melt season. Ice surface temperature was calculated from outgoing longwave radiation (Table 1) using

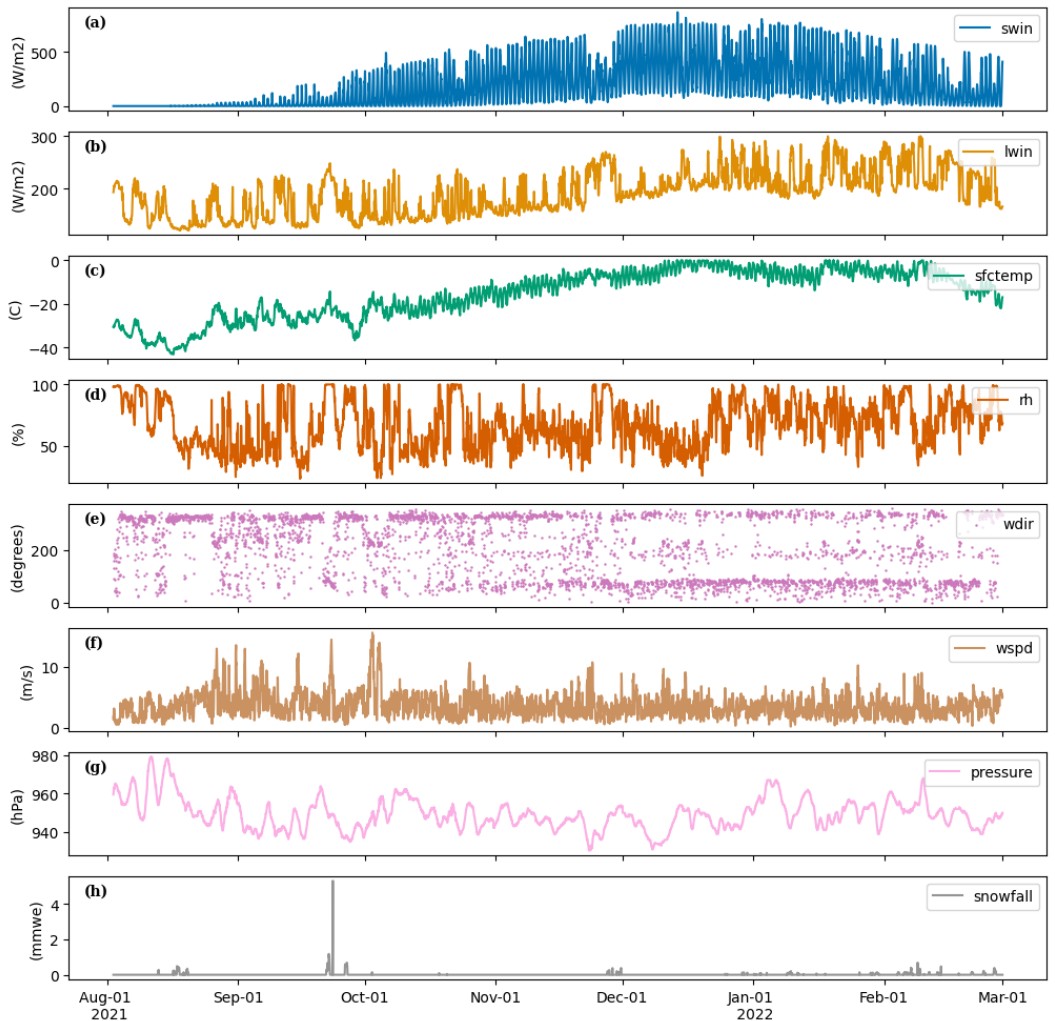

**Figure 3.** (a) Average hourly incoming shortwave radiation, incoming longwave radiation, surface temperature, relative humidity, wind direction, wind speed, surface pressure and snowfall measured at COHM AWS (1 August to 30 November 2021) and CWG AWS (1 December 2021 to 28 February 2022). Pressure is from the Antarctic Mesoscale WRF Prediction System (AMPS) for the first period as it was not measured at COHM AWS.

the Stefan-Boltzmann law and an emissivity of 1. Ice temperatures were obtained from thermocouples deployed at six different depths (Table 1). A two week validation period in early December was chosen as the top two thermocouple sensors melted out after the first two weeks of December. Observed albedo was calculated as accumulated albedo, by taking a moving mean over 24 hours for the ratio of accumulated outgoing shortwave radiation (absolute value) over accumulated incoming shortwave radiation (Van Den Broeke et al., 2004). This method is used in polar regions to limit uncertainty associated with changing solar zenith angle with the midnight sun. As noted in Section 3.3, surface height change was obtained using the K nearest

220

neighbors regression used for pre-processing precipitation to eliminate noise from the sensor. Finally, streamflow data came from the Lost Seal stream gauge (Gooseff and Mcknight, 2021) and were used to validate the temporal variation in simulated runoff.

## 4 Modifications

This section describes the two different schemes in Crocus that required modification: vertical percolation of water through the snowpack and spectral albedo. We describe the processes accounted for in the model, the challenges these schemes pose for a cold-based MDV glacier and a description of the modifications made.

### 4.1 Water drainage through snowpack/glacier

Eidhammer et al. (2021) implemented WRF-Hydro/Glacier on a temperate glacier in Norway with internal temperatures initialized at 0 °C. Runoff in the summer is generated by precipitation (rain) in addition to melt. In contrast, the MDV glaciers are cold-based with internal temperatures around -18 °C and runoff is entirely from surface and near-surface glacial melt. Due to the extreme differences in these two environments, modifications to the snowpack/ice water flow were necessary in order to adapt Crocus embedded in WRF-Hydro/Glacier to this environment.

As a snowpack model, runoff in the original WRF-Hydro/Glacier is calculated as the remaining water after it has percolated through all of the layers and reached the ice-soil interface. In this version of the model, ice is not treated differently to snow layers and the ice remains porous. This is problematic as it allows meltwater to percolate through the ice layers rather than having the ice layers act as a barrier (MacDonell, 2008; Bergstrom et al., 2021). As the water percolates, it can refreeze in a layer depending on temperature and energy or it can be held in the layer as liquid water content. Fig. 4a depicts the water flow scheme from WRF-Hydro/Glacier (oldrunoff) for a single hour in the top 40 cm or 4 layers of the glacier. Fig. 4a shows that meltwater is generated in the first layer and a portion of that melt is held in the layer as liquid water content since it is a snow layer. The remaining meltwater percolates down to the second layer, which is an ice layer. Some of the percolated water refreezes and a smaller amount is held as liquid water content of the layer. Any remaining water continues to percolate to the third layer, where all of it refreezes and thus, there is no runoff.

Runoff is only possible in oldrunoff if the entire glacier is at the melting point such that the liquid water can percolate through the entire glacier or snowpack column. This does not align with MDV observations of near-surface melt with steep subsurface temperature gradients (Fountain et al., 1998; Hoffman et al., 2014). Thus, we introduce a condition such that if the liquid water reaches an ice layer, a portion refreezes and all remaining water becomes runoff. This allows the ice to be a barrier and results in the generation of near-surface runoff. The effect of this modification can be seen in Fig. 4b, newrunoff. Here, the top snow layer is similar to Fig. 4a and a portion of the water that percolates through the top snow layer refreezes on the second layer. The difference is that there is no liquid water held in the second layer, since the ice layers are no longer porous and have a

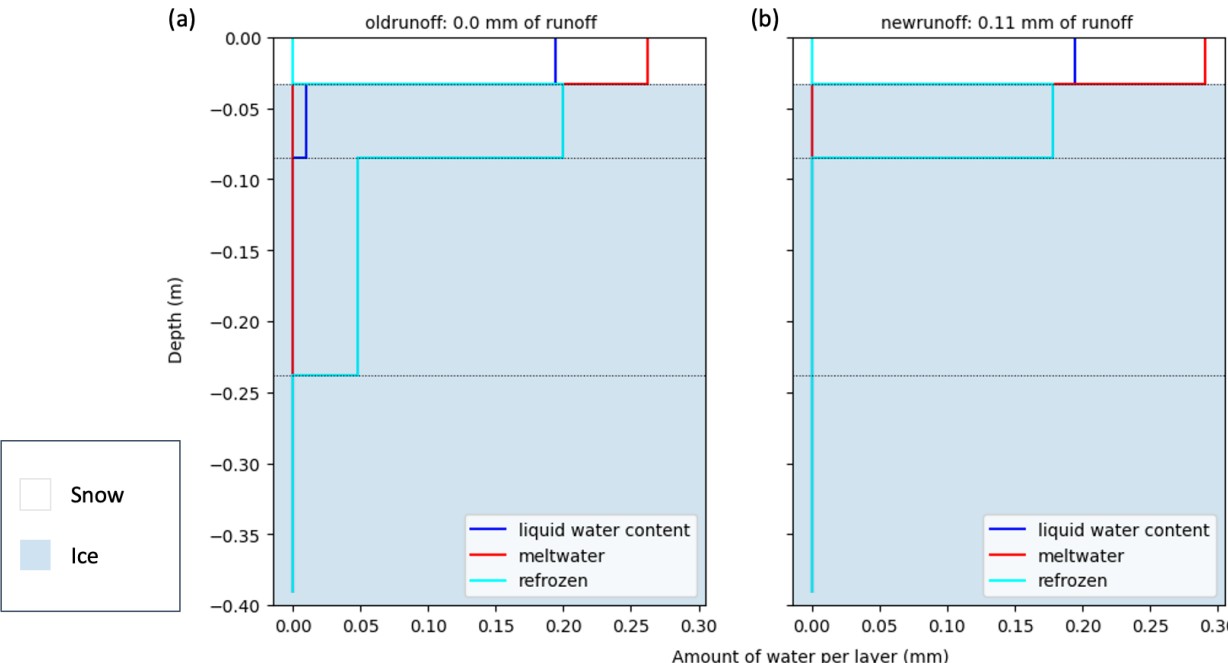

**Figure 4.** A comparison between the (a) original runoff scheme from Crocus (oldrunoff) and the (b) new modified runoff scheme (newrunoff). Cross sections show the top 40 cm and the total amount of liquid water held in the layer (blue), the total amount of water melted (red) and total amount refrozen (cyan) for each layer at 3:00 UTC 10 December 2021. The white and blue background colors show which layers are snow and ice, respectively. Horizontal black dotted lines denote the different layers.

liquid water holding capacity of zero. The remaining liquid water from the second layer becomes runoff since the ice layer is a barrier preventing further percolation. Thus, there is 0.11 mm of runoff in newrunoff.

## 4.2 Crocus spectral albedo scheme in WRF-Hydro/Glacier

In this section, (1) we describe how broadband albedo is calculated from spectral albedo in WRF-Hydro/Glacier, (2) provide the rationale for modifying the parameters in the albedo scheme and (3) show the new parameters tuned to observations of spectral and broadband albedo.

### 4.2.1 Description of albedo scheme

The Crocus albedo scheme implemented in WRF-Hydro/Glacier is currently calculated over three spectral bands, where Band 1 is the visible band with wavelengths from 0.3-0.8 $\mu$m, Band 2 is the red band with wavelengths from 0.8-1.5 $\mu$m and Band

3 is the near infrared band with wavelengths from 1.5-2.8 $\mu$m. The albedo in each band is calculated for the top two layers of the glacier using different schemes for ice and snow.

The Crocus snow albedo scheme in WRF-Hydro/Glacier is based on the theoretical studies of Warren (1982). Albedo for each band ($i = 1..3$) of the two top layers ($j = 1, 2$) is calculated in the following equations:

$$
\begin{cases}
\alpha_1^j &= \max\left(S_{1a}, \min\left(S_{1b}, S_{1c} - S_{1d}\sqrt{d_{\text{opt}}^j}\right) - \min\left(1, \max\left(\frac{P}{870}, 1.5\right)\right) \times 0.2\frac{A^j}{60}\right) \\
\alpha_2^j &= \max\left(S_{2a}, S_{2b} - S_{2c}\sqrt{d_{\text{opt}}^j}\right) \\
\alpha_3^j &= S_{3a}d'^j - S_{3b}\sqrt{d'^j} + S_{3c},
\end{cases}
\tag{1}
$$

where $S$ are various snow model parameters (see Table 3), $d_{\text{opt}}$ is the optical diameter of the snow grains, $P$ is the mean surface pressure in hectopascals, $A$ is the snow age in days and $d'$ is the minimum of $d_{\text{opt}}^j$ and 0.0023 for each layer, $j$. All of the constants in the three equations correspond to parameters that can be altered in the Crocus namelist (see Table 3).

For ice layers (density $> 850$ kg m$^{-3}$), the albedo of the three bands ($\alpha_1, \alpha_2, \alpha_3$) is:

$$
\alpha_i^j = I_i, \quad i = 1..3, j = 1, 2
\tag{2}
$$

where $I_i$ are the constant values for ice for the three bands ($i$) listed in Table 3.

Next, a weight ($f$) is calculated for both layers as a function of the thickness of each layer and various parameters:

$$
f^1 = 0.8 \times \min\left(1, \frac{\Delta z^1}{0.02}\right) + 0.2 \times \min\left(1, \max\left(0, \frac{\Delta z^1 - 0.02}{0.01}\right)\right)
\tag{3}
$$

$$
f^2 = 0.8\left(1 - \min\left(1, \frac{\Delta z^1}{0.02}\right)\right) + 0.2\left(1 - \min\left(1, \max\left(0, \frac{\Delta z^1 - 0.02}{0.01}\right)\right)\right),
\tag{4}
$$

where $\Delta z^1$ is the thickness of the first layer. The total albedo of each band ($\alpha_i$), is then

$$
\alpha_i = f^1\alpha_i^1 + f^2\alpha_i^2, \quad i = 1..3
\tag{5}
$$

and the broadband albedo is a weighted average of the total albedo of each band:

$$
\alpha = 0.71\alpha_1 + 0.21\alpha_2 + 0.08\alpha_3.
\tag{6}
$$

### 4.2.2 Motivation for tuning the parameters

Fig. 5 shows the observed broadband albedo from 1 December to 28 February. There was snowfall at the end of November (Fig. 3h) so on 1 December the timeseries begins with snow that is no longer fresh. The surface has a broadband albedo of 0.7, then the surface transitions from snow to ice on 20 December with an albedo of 0.55. After, there are a few snowfall events

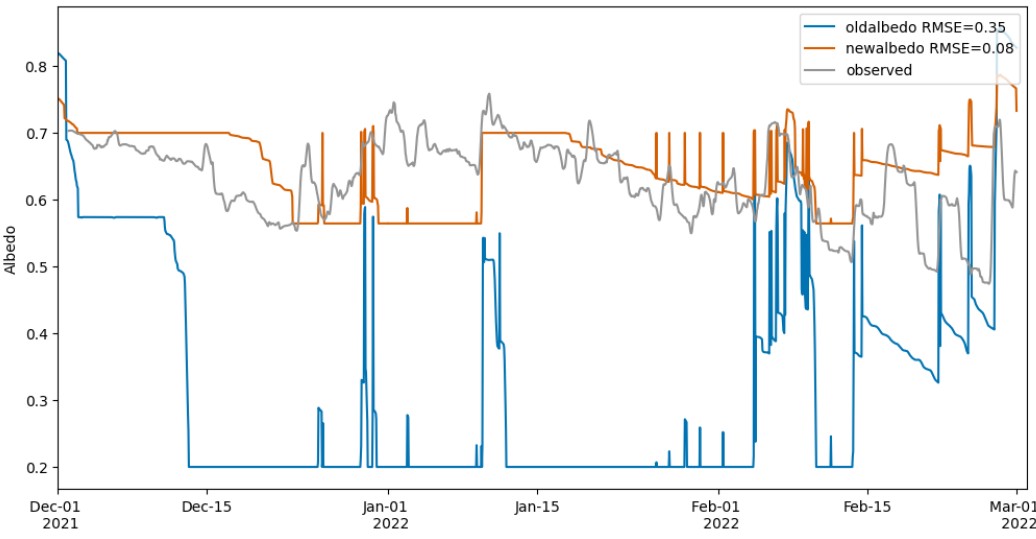

**Figure 5.** Comparison between the broadband albedo of the original Crocus scheme (oldalbedo) in blue, the modified scheme (newalbedo) in orange and the observed daily accumulated albedo in grey.

(e.g. 24, 26, 30 December and 5, 11 January) that increase albedo followed by a slow decay after each snowfall. Looking at modelled albedo, oldalbedo (Fig. 5) begins with a snow albedo of 0.81 then the surface transitions to ice with a constant albedo of 0.2 on 13 December. Each time it snows, the albedo increases briefly, then rapidly decays to ice.

The issues are 1) the modelled ice albedo is too low compared to the observations (0.2 vs. 0.55) and 2) the modelled snow albedo does not match the observations (0.7 vs 0.81). The observed ice albedo only lowers to 0.48 and is similar to the range of minimum albedo measured by Bergstrom et al. (2020) of approximately 0.4-0.5 on Commonwealth Glacier over three melt seasons. Hoffman (2011) similarly found a minimum of ∼0.53. These values are higher than temperate glaciers because of minimal sediment and impurities on the glacier and low amounts of meltwater present that can lower albedo (Hoffman, 2011). The snow albedo observations at the beginning of the period are lower than the modelled snow albedo. Observed snow albedo over the melt season is also similar to the albedo observations from Bergstrom et al. (2020), however the values for fresh snow are often lower than those observed in other glaciated locations. For example, Reijmer et al. (2001) recorded an average snow albedo of 0.78 at a point in Dronning Maudland, Antarctica, whilst the average snow albedo in this study is ∼0.65. This is likely because there are very little snowfall amounts in the MDV and the albedo will be a combination of thin surface snow and the lower albedo of the ice beneath. The parameters in the model snow albedo scheme are based on modelled snow albedo in Warren (1982) and the parameters are optimised for the French Alps (Brun et al., 1992). The ice albedo scheme is constant and the parameters have been tuned to glaciers in the French Alps (Gerbaux et al., 2005). Thus, the parameters of the model

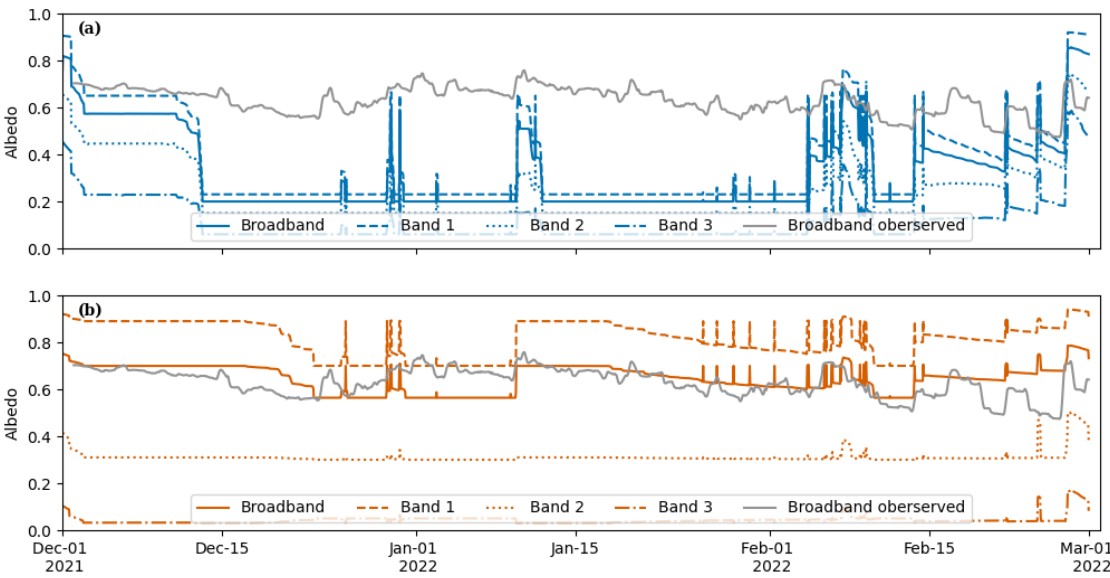

**Figure 6.** Comparison between the daily accumulated broadband albedo in grey and three spectral albedo bands of (a) the original Crocus albedo (oldalbedo) in blue and (b) the modified scheme (newalbedo) in orange.

albedo scheme must be modified to better represent the conditions found on the glaciers in the MDV.

### 4.2.3 Modifying the albedo scheme parameters

We chose to tune the albedo scheme for December 2021 and test the new parameters from 1 January to 28 February 2022. In this study, we opted to use observed spectral albedo profiles measured by Dadic et al. (2013) in Allan Hills, East Antarctica as they provide similar information to Warren (1982), but for a location that is closer to our field site on Commonwealth Glacier.

First, we determine new parameters for the ice albedo bands by calculating the average spectral albedo of each band from Dadic et al. (2013, Fig. 6) for the different types of ice (white and blue ice). These values were then tuned to the minimum observed broadband albedo during December 2021, whilst ensuring that the value of each band remained in the range of average values from Dadic et al. (2013). The effect of this tuning process for ice albedo is shown from 13-25 December in Fig. 5, where newalbedo is much closer to the broadband albedo observed on the ice surface in this period.

**Table 3.** Parameters from original Crocus in WRF-Hydro/Glacier (oldalbedo) compared to the new scheme (newalbedo). Bold indicates changed parameters. We introduce a new parameter, XVALB12 ($S_{2a}$) to specify the minimum albedo for Band 2 as the original value was hard coded.

| Snow albedo | | | | |
|---|---|---|---|---|
| | Parameter | oldalbedo | newalbedo | Name in code |
| Band 1 | $S_{1a}$ | 0.65 | **0.89** | XVALB11 |
| | $S_{1b}$ | 0.92 | **0.94** | XVALB4 |
| | $S_{1c}$ | 0.96 | 0.96 | XVALB2 |
| Band 2 | $S_{2a}$ | 0.3 | **0.31** | XVALB12 (new parameter) |
| | $S_{2b}$ | 0.9 | **0.66** | XVALB5 |
| | $S_{2c}$ | 15.4 | 15.4 | XVALB6 |
| Band 3 | $S_{3a}$ | 346.3 | 346.3 | XVALB7 |
| | $S_{3b}$ | 32.31 | **21.0** | XVALB8 |
| | $S_{3c}$ | 0.88 | **0.35** | XVALB9 |
| Ice albedo | | | | |
| Band 1 | $I_1$ | 0.23 | **0.7** | XALBICE1 |
| Band 2 | $I_2$ | 0.15 | **0.3** | XALBICE2 |
| Band 3 | $I_3$ | 0.06 | **0.05** | XALBICE3 |

Next, we modified the maximum and minimum spectral albedo in each band based on observations by Dadic et al. (2013) (Band 1 are $S_{1b}$ and $S_{1a}$, respectively and Band 2, $S_{2b}$ and $S_{2a}$). Band 3 is slightly more complex. Here, we solved a system of equations using the largest and smallest optical diameters from the oldalbedo to represent snow and firn over the tuning period. These maxima and minima were constrained by averages calculated from Fig. 6 in Dadic et al. (2013) for snow and firn and tuned to observed broadband albedo ($S_{3b}$ and $S_{3c}$). The new parameter choices are in bold in Table 3. Fig. 6b shows the effect of the new parameters on the three bands and the broadband albedo over the full melt season. Bands are separate in newalbedo and are aligned with theory, as well as observations from Dadic et al. (2013). Variability in observed albedo is better captured by newalbedo over the melt season with a root mean square error of 0.08 compared to oldalbedo with a root mean square error of 0.35. Accurately simulating albedo enables us to better simulate the feedbacks between albedo, precipitation and melt.

## 5 Model comparison and evaluation

In this section, we evaluate three different versions of the model code representing the original scheme (oldrunoff_oldalbedo), updates to both runoff and albedo schemes (newrunoff_newalbedo) and updates only to the runoff scheme (newrunoff_oldalbedo). The versions are evaluated against observations of surface temperature, ice temperatures and surface height change over summer 2021/22. Finally, the differences in simulated runoff between the three models are presented and discussed.

### 5.1 Surface temperature

Fig. 7 shows the performance of the different models compared to the observed surface temperatures at the AWS over the second half of December. The timeseries shows that the newrunoff_newalbedo daytime surface temperature drops below the melting point from 26-30 December similarly to observed surface temperatures, while the two previous versions of the model (oldrunoff_oldalbedo and newrunoff_oldalbedo) reach the melting point daily during that period. This is more evident in Fig. 8 where the two models using the original albedo scheme have similar root mean square errors (Fig. 8a, b), however the root mean

square error of newrunoff_newalbedo (Fig. 8c) is 60% less than newrunoff_oldalbedo and 62% less than oldrunoff_oldalbedo. Furthermore, newrunoff_newalbedo has a more equal spread on either side of the 1-1 line around the melting point compared to the two models with the original albedo scheme, which both overestimate the number of hours with surface temperature at the melting point as seen by the aggregation of points on the top left of the 1-1 line (Fig. 8a,b). The percentage of time (hours) each model is at the melting point is 25.8 % (557.3 hours) for oldrunoff_oldalbedo, 24.5 % (529.2 hours) for newrunoff_oldalbedo

and 5.7 % (123.12 hours) for newrunoff_newalbedo, compared to 5.0 % (108.0 hours) for observed surface temperatures. In comparison, Hoffman et al. (2014) found that on average there were 255.1 hours of surface and near-surface melt with a maximum of 452 and a minimum of 92 hours from 1996-2009 on Taylor Glacier. Thus, the values for newrunoff_newalbedo are similar to the average, while oldrunoff_oldalbedo and newrunoff_oldalbedo are greater than the maximum number of hours found by Hoffman et al. (2014).


For temperatures around -20 °C, newrunoff_newalbedo has a cold bias of up to 2 °C, which can also be seen in the daily minimums in Fig. 7. This is likely due to the observed albedo being lower than the modelled albedo during this period as seen in Fig. 5.

### 5.2 Internal ice temperatures

Internal ice temperatures at six different depths are validated for the three model runs in Table 4. This shows that newrunoff_newalbedo has the lowest root mean square errors for internal ice temperatures at TC1-4 and 6. Fig. 9 shows the temporal variability of the modelled vs observed ice temperatures for newrunoff_newalbedo. The modelled temperatures agree with the measured temperatures and are often within 1 °C of the measurement. The root mean square errors for TC2-6 all lie below one except for at TC1, which matches observations at the beginning of the period, but has a cold bias and a phase shift later in the period. The

first time step of the period beginning 1 December shows that the model is in good agreement with the observations and gives

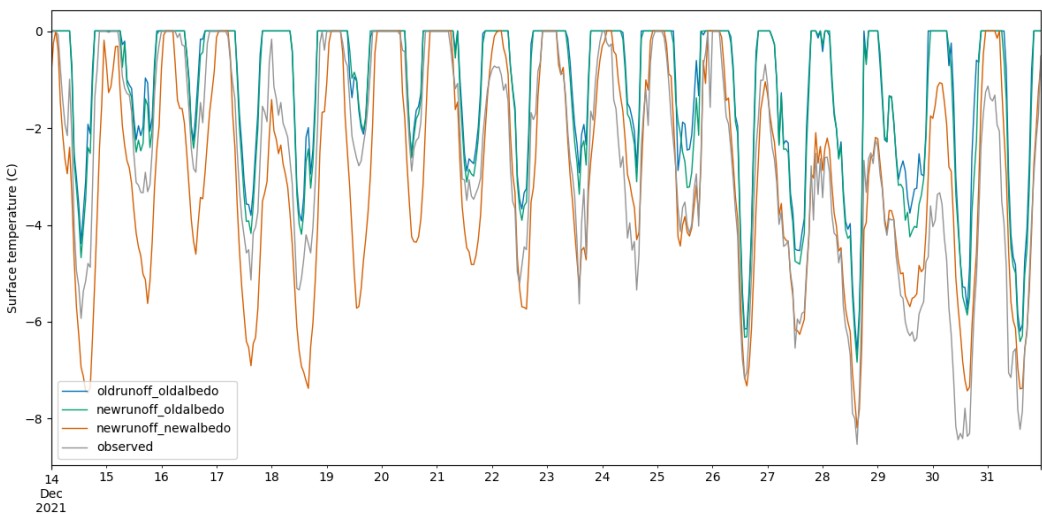

**Figure 7.** Comparison between the three different versions of the model and the observed surface temperatures over the last two weeks in December 2021.

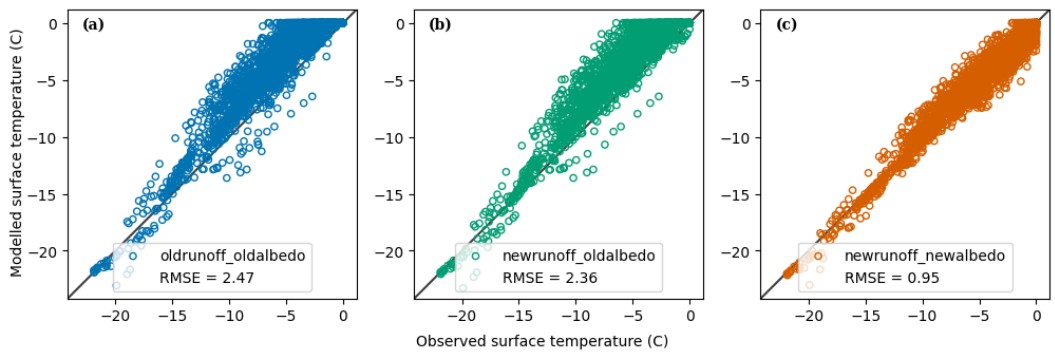

**Figure 8.** A scatter plot with modelled surface temperatures vs. observed surface temperatures for the three models from 1 December 2021 to 28 February 2022.

us confidence that a longer spin-up is not necessary. The model also captures both the diurnal cycle and the seasonal warming seen in TC4, TC5 and TC6. This can also be seen in Fig. 10, which shows a cross section of ice temperatures in the top 3 meters of the glacier modelled by newrunoff_newalbedo from August 2021 through February 2022. Darker blues indicate the cooling of the glacier during the spin-up period (August-November 2021) and then the warming period beginning in December.

However, there is a slight delay in the diurnal cycle of the measurements compared to the model in Fig. 9. This could be due to differences in solar penetrating radiation that result in the glacier not warming enough compared to the measurements. Overall,

**Table 4.** Root mean square errors of the internal ice temperatures of the three model versions at six different depths from 1-15 December 2021.

|  | oldrunoff_oldalbedo | newrunoff_oldalbedo | newrunoff_newalbedo |
|---|---|---|---|
| TC1 | 2.36 | 2.27 | 1.58 |
| TC2 | 2.26 | 1.91 | 0.9 |
| TC3 | 2.63 | 1.96 | 0.53 |
| TC4 | 2.24 | 0.7 | 0.56 |
| TC5 | 1.25 | 0.66 | 0.88 |
| TC6 | 0.28 | 0.28 | 0.27 |

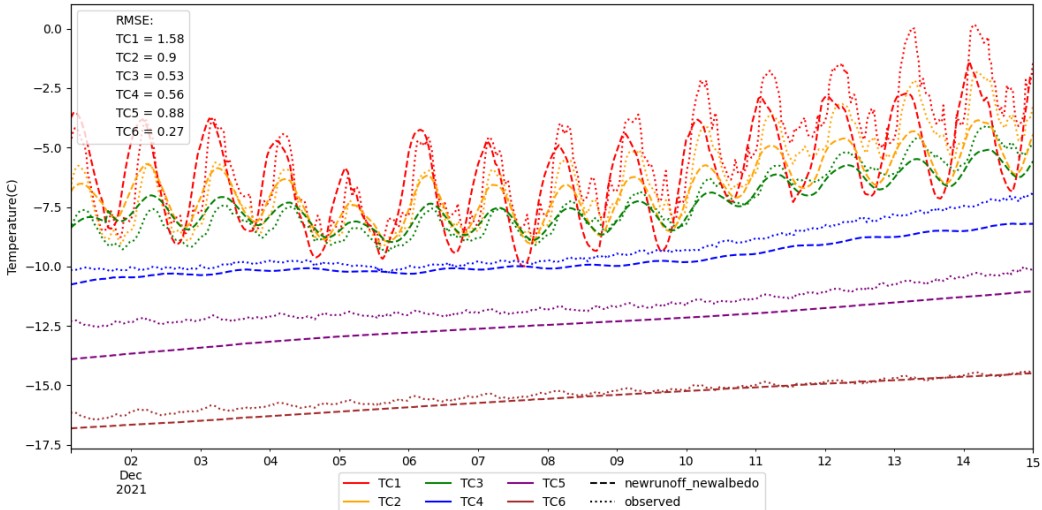

**Figure 9.** Modelled internal ice temperatures from newrunoff_newalbedo for 1-15 December 2021. Thermocouples were initially deployed at six different depths: 0.05, 0.1, 0.2, 0.5, 1.0 and 2.0 meters. Note that the measurements are not at a constant depth with time as the sensors were melting out over the period. This shows diurnal variability at shallower depths and the seasonal warming wave below.

the model does a satisfactory job of modelling the near-surface ice temperatures as seen in the root mean square errors in Fig. 9, which is important for generating near-surface melt.

## 5.3 Surface height change

Fig. 11 shows the surface height change from 1 December 2021 to 28 February 2022 of the three model versions compared with observations of surface height. Although the three models begin December in alignment (same spin-up), they begin to diverge a couple days later. The model that is most similar to the observed surface height is newrunoff_newalbedo with a

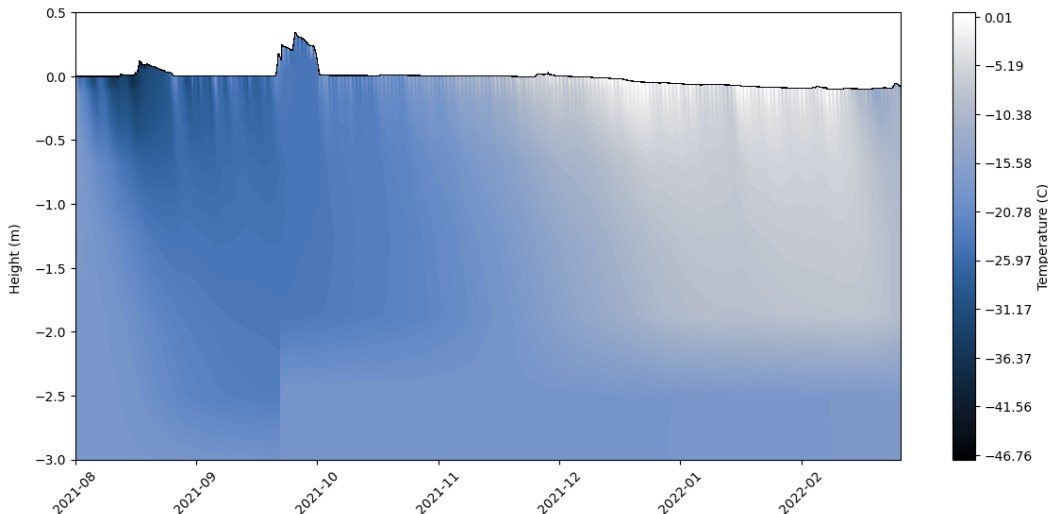

**Figure 10.** Cross section of modelled temperature from newrunoff_newalbedo of the top three meters of the glacier over the spin-up period of August-November and validation period of December-February. This shows the diurnal variability at shallower depths and the seasonal cooling then warming wave below.

root mean square error of 0.02. The model that performs the worst is oldrunoff_oldalbedo with a 35 cm more decrease in surface height than the observations over the three month period and a root mean square error of 0.26. The performance of
newrunoff_oldalbedo is similar to newrunoff_newalbedo, but with a root mean square error of 0.05.

The differences between the surface height change of the three models occur during the first three weeks of December. After 23 December, the rate of decrease in the change in surface height across the three models stabilizes. From this point to the end of the simulation, oldrunoff_oldalbedo has 90 cm ablation, while newrunoff_oldalbedo has 10 cm ablation and
newrunoff_newalbedo has 3 cm ablation compared to 10 cm observed ablation. On 6 December, oldrunoff_oldalbedo and newrunoff_oldalbedo reach the melting point for the first time and the suface height lowers faster than newrunoff_newalbedo. On 13 December, TC1-3 near-surface ice temperatures of oldrunoff_oldalbedo reach the melting point (Fig. A1), whilst newrunoff_oldalbedo near-surface ice temperatures remain below the melting point (Fig. A2). This rapid decrease in surface height and warming of the glacier layers for oldrunoff_oldalbedo may be due to near-surface melt that is percolating through ice
layers and subsequently warming them, rather than draining as near-surface runoff as modified in newrunoff_oldalbedo. This can be seen in Fig. A1 for oldrunoff_oldalbedo, where TC1 reaches the melting point on 12 December, followed by internal ice temperatures for TC2-5 increasing rapidly from 13 December. On the other hand, in Fig. A2, TC1 from newrunoff_oldalbedo reaches the melting point on 13 December, but TC2-5 do not increase as much as in oldrunoff_oldalbedo. Thus, it is evident that optimizing the parameters for the albedo scheme and modifying the drainage through the glacier was needed to accurately
simulate surface height change.

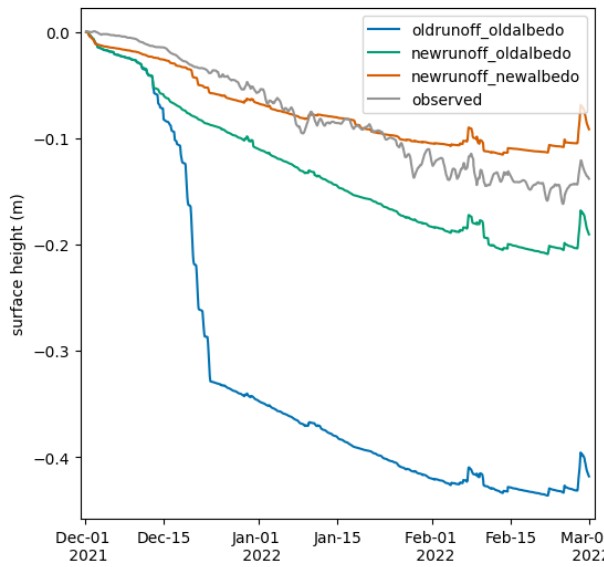

**Figure 11.** The change in surface height (m) between the three models and the quarter-hourly observational data.

## 5.4 Runoff

Although we do not have site specific runoff measurements for this location, stream gauge observations of the Lost Seal catchment on Commonwealth Glacier (Gooseff and Mcknight, 2021) indicate that melt and runoff are activated episodically, rather than consistently melting daily over the melt season (Fig. 12c). Fig. 12a shows the daily accumulated runoff in millimeters of liquid water for the three different models. There is no runoff from oldrunoff_oldalbedo during the entire period of the melt season due to the percolation and subsequent refreezing of meltwater through the ice layers that was modified for the newrunoff models. In comparison, from 10 December runoff is generated almost daily throughout the season in newrunoff_oldalbedo, with a daily maximum of 45.0 mm whereas runoff is more episodic in newrunoff_newalbedo, with a daily maximum of 5.1 mm.

Fig. 12a, b show the runoff is generated earlier and is larger in newrunoff_oldalbedo than newrunoff_newalbedo. This is because the albedo falls to the ice albedo on the day that runoff is activated for the season (Fig. 5). There is less runoff overall in newrunoff_newalbedo and runoff drops to zero frequently compared to newrunoff_oldalbedo, which has runoff every day from the start to the end of the season. When we compare the runoff with surface temperatures from Fig. 7, it can be seen that runoff is activated when ice surface temperatures reach the melting point and the runoff shuts off when the temperature drops below the melting point.

Furthermore, we can compare the daily average modelled runoff from newrunoff_newalbedo to the daily average stream gauge observations (Gooseff and Mcknight, 2021) in Fig. 12c. Although we cannot compare the magnitudes, this shows that we

are accurately capturing the temporal variability and episodic nature of melt (Fig. 12c). Modelled runoff is not accounting for the physical processes of water drainage off-glacier, such as evaporation or soil absorption and this may explain some of the differences between the stream gauge and model. Hoffman et al. (2014) calculated a range of 11.2 to 152.0 mm water-equivalent (w.e.) of total subsurface drainage and surface melt per melt season over 13 years at a point on Taylor Glacier. The total runoff over the melt season in our study was 0 mm w.e. for oldrunoff_oldalbedo, 858 mm w.e. for newrunoff_oldalbedo, and 36 mm w.e. for newrunoff_newalbedo. The total runoff for newrunoff_newalbedo falls in the range from Hoffman et al. (2014). Thus, we are confident that newrunoff_newalbedo is modelling the frequency and magnitude runoff accurately.

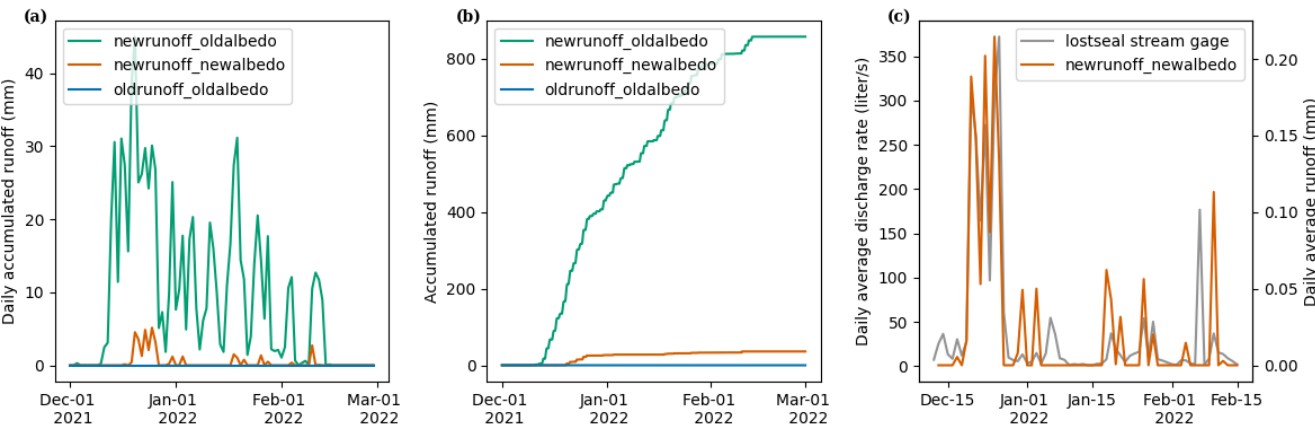

**Figure 12.** (a) Daily accumulated runoff and (b) cumulative runoff from the three model runs from 1 December 2021 - 28 February 2022. (c) Daily average discharge from the Lost Seal stream gauge (grey) compared to modelled daily average runoff (orange) from newrunoff_newalbedo.

## 5.5 Limitations

We implemented the WRF-Hydro/Glacier model at a point on Commonwealth Glacier and forced the model with observational data in order to limit the input uncertainties to the model that would be introduced if using atmospheric model or gridded data. However, a few uncertainties in the input data still exist, especially regarding precipitation. Precipitation was calculated from the change in snow height record, but this was challenging as the data are noisy and summer snow accumulation is very small, increases albedo and pauses melt generation (Fountain et al., 2010). Temperature and wind speed can impact the sonic ranging sensor that measures the snow height and contribute to the noise in the data. The threshold for snow was tuned on a monthly basis and was smaller than both the instrument accuracy and the thresholds from Fountain et al. (2010) and Myers et al. (2022), which would have filtered out all of the summer snowfall events seen in observed albedo.

Other limitations to this study are the short time period used for spin-up, tuning and testing of WRF-Hydro/Glacier. We demonstrate that the four month spin-up is sufficient to accurately simulate the internal ice temperatures. Although a longer

time period could be helpful for tuning and testing the model, this study is a proof of concept and the evidence provided demonstrates that one month of tuning is sufficient to simulate the evolution of the broadband albedo for January and February.


## 6 Future research and outlook

To better understand the physical processes governing the hydrological cycle in the MDV it is necessary to resolve the connections between the atmosphere, glaciers, bare-land surfaces and soils, stream channels and lakes (hydrological reservoirs). What makes the MDV unique compared to other glaciated environments is that changes in surface energy balance are the primary

control on melt water generation and mass balance, as precipitation is limited. Although previous studies have implemented surface energy and mass balance models on glaciers in the MDV glaciers (Hoffman, 2011; Macdonell et al., 2013; Hofsteenge et al., 2022), they have not been able to fully account for the meltwater pathways from the glaciers to the surrounding landscape. By implementing a new modelling framework that couples a detailed snowpack model to a fully distributed hydrological model (WRF-Hydro/Glacier), we have taken the first step in enabling the hydrological connectivity of the MDV to be further assessed.


To successfully simulate the onset, duration and end of melt on a cold-based glacier in the MDV, the multi-layer snowpack scheme in WRF-Hydro/Glacier needed to be optimised, which was achieved at a point scale over a 7-month period encapsulating a melt season. The snowpack model (Crocus), which is embedded in WRF-Hydro/Glacier, was (1) modified to limit the percolation of meltwater in the presence of ice layers and (2) optimised to improve the parameter set controlling albedo

and net shortwave radiation. We demonstrate that simulating albedo, which has not been attempted before on a glacier in the MDV, is necessary to resolve the complex feedbacks between albedo, snowfall and melt in this energy limited environment. Our approach to simulate albedo based on the evolution of snow grain properties is a significant step forward for modelling glacier response to climate change in the MDV compared to using point-based observations of albedo (e.g. Hoffman (2011); Hofsteenge et al. (2022)).


Future research will be able to utilise this modelling framework to better resolve the spatial and temporal variability in albedo, which is critical in governing spatially distributed melt and hydrological connectivity in the MDV. For example, Bergstrom et al. (2020) measured the spatial variability of albedo from a series of radiometric observations obtained from helicopter flights over three MDV glaciers, which showed that albedo not only increased with elevation but also increased from west to

east across both the Canada and Commonwealth glaciers in the Taylor Valley. The longitudinal patterns in albedo observed by Bergstrom et al. (2020) demonstrate that point-based observations of albedo are not sufficient to resolve the spatial and temporal variability in albedo. Thus, the optimised simulation of albedo in the multi-layer snowpack scheme in WRF-Hydro/Glacier used in this study provides a new pathway to resolve this observed complexity, which is critical in governing the amount of meltwater generated from glaciers in the MDV. Importantly, it provides us with confidence that we have developed a modelling

framework that will enable us to get the "right answers for the right reasons" (Kirchner, 2006) in regard to resolving one of the

key physical processes governing the spatial variability of melt.

The modified WRF-Hydro/Glacier model will also allow us to expand on the work from Hoffman (2011) and Cross et al. (2022) by accounting for in-stream processes such as evaporation and soil absorption of meltwater. These processes are expected to have the greatest impact on streamflow in low melt years and in the larger, more complex tributaries. Furthermore, explicitly modelling the stream channels will allow us to answer questions about the timing between melt generation and lake inflow, which has downstream impacts on nutrient availability for the microbial ecosystems found in the MDV (Gooseff et al., 2017; Singley et al., 2021). By having the ability to better understand streamflow dynamics and hydrological connectivity, it is anticipated future studies using this modelling framework will be capable of providing new insights into the impacts of climate forcing on meltwater generation.

*Code and data availability.* Code of the modified WRF-Hydro/Glacier for a cold-based glacier is available on github (https://github.com/tpletzer/wrf_hydro_nwm_coldglacier).

*Author contributions.* TP, JPC, NJC and MK designed the study. TP and JPC devised the methodology. TP implemented and modified WRF-Hydro/Glacier, collected CWG AWS data and produced the results. TE provided expert advice on the WRF-Hydro/Glacier model. TP, JPC and NJC contributed to the analysis. TP wrote the manuscript with edits from the co-authors. MK provided funding for the field work.

*Competing interests.* The contact author has declared that they do not have any competing interests.

*Acknowledgements.* This research has been supported by the Antarctic Science Platform (grant no. ANTA1801) and the Royal Society of New Zealand (grant no. RDF-UOC1701). We would also like to thank University of Otago for funding this research. The authors wish to acknowledge the use of New Zealand eScience Infrastructure (NeSI) high performance computing facilities and consulting support as part of this research. New Zealand's national facilities are provided by NeSI and funded jointly by NeSI's collaborator institutions and through the Ministry of Business, Innovation and Employment's Research Infrastructure programme https://www.nesi.org.nz.

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

# Appendix A: Internal ice temperatures from oldrunoff_oldalbedo and newrunoff_oldalbedo

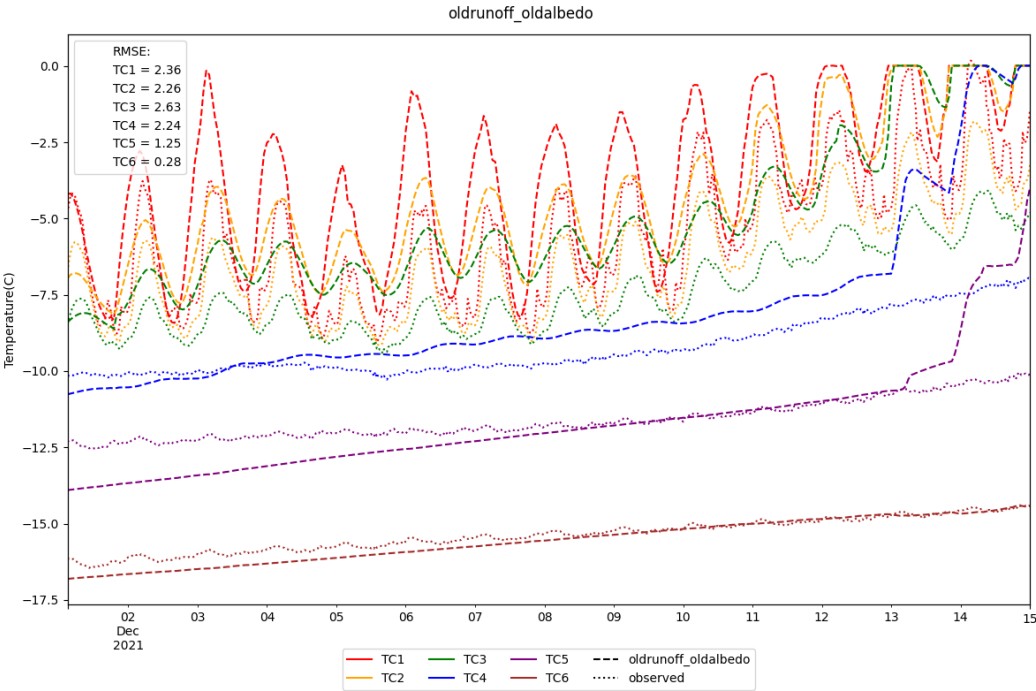

**Figure A1.** Modelled internal ice temperatures from oldrunoff_oldalbedo over the first two weeks of December. Thermocouples were initially deployed at six different depths: 0.05, 0.1, 0.2, 0.5, 1.0 and 2.0 meters. Note that the measurements are not at a constant depth with time as the sensors were melting out over the period.

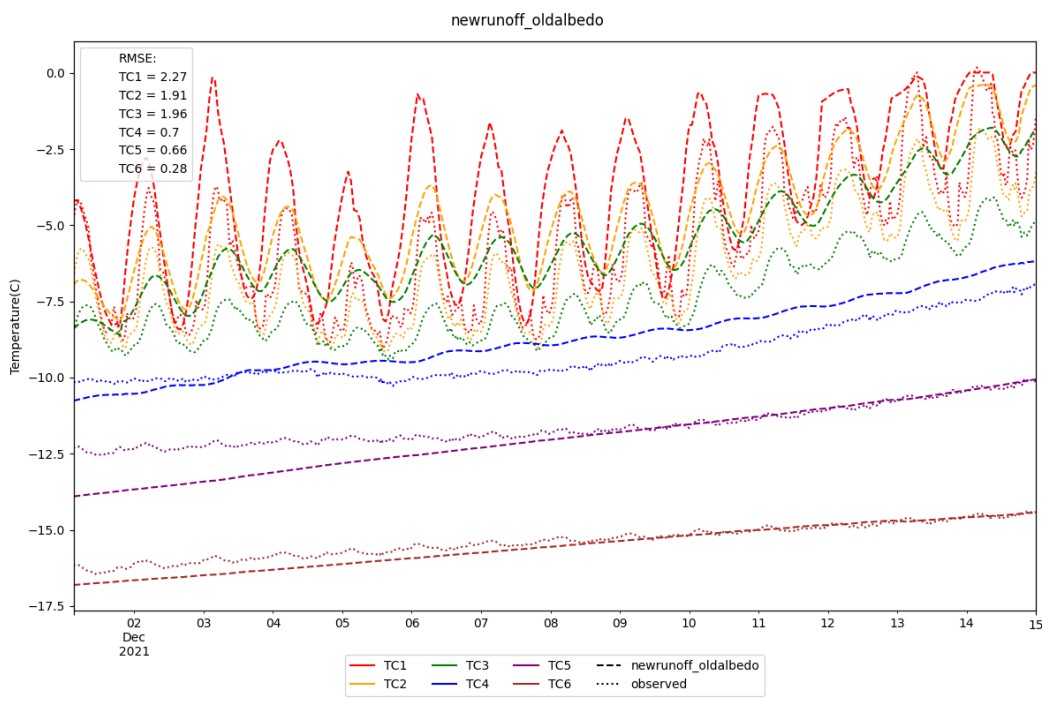

**Figure A2.** Modelled internal ice temperatures from newrunoff_oldalbedo over the first two weeks of December. Thermocouples were initially deployed at six different depths: 0.05, 0.1, 0.2, 0.5, 1.0 and 2.0 meters. Note that the measurements are not at a constant depth with time as the sensors were melting out over the period.