# Peer review of "The application and modification of WRF-Hydro/Glacier to a cold-based Antarctic glacier"

_EGUsphere, 2023_

## Author Comment (AC1)

**Response to reviewer comment #1**. Note reviewers' text is shown in blue, with responses in **black**.

General comments:

Pletzer at al. present a point-based optimization and evaluation of a simplified version of the WRF-Hydro/Glacier modelling system over the McMurdo Dry Valleys (MDVs) using forcing data from automatic weather stations. They identify two aspects of the cryospheric component needing improvement for accurate simulations of runoff, namely the representation of percolation in ice layers and the parameters in the albedo scheme. The manuscript is well-written and organized, the results are clearly and concisely presented, and the topic suits the scope of the journal. However, I have a concern about the novelty and wider applicability of the presented results due to the simplified model configuration employed, as outlined in my major comment below, which should be considered prior to publication.

We express our gratitude for taking the time to review our manuscript and offering valuable feedback for its improvement. We have taken your comments into consideration and made significant revisions. Specifically, we have reworked the introduction that includes a new and much clearer research aim, and conclusion to explicitly highlight the scientific advancement of our research. To eliminate any confusion regarding the motivation for the analysis, we have added a schematic of the WRF-Hydro/Glacier modelling framework. Additionally, we have addressed the minor comments provided. Thank you once again for your time and insightful input.

Major comment:

As I understood, the authors only used the cryospheric (Crocus) and land-surface (Noah-MP) modeling components from WRF-Hydro/Glacier. The presented work is therefore mainly small changes to/calibration of parts of Crocus, which is an important foundational step for tackling interesting science questions in a new region but is itself a methodological task. As a result, the introduction lacks a clear scientific question and, in my opinion, the results may be insufficiently novel, as previous studies have applied Crocus in Antarctica (including in coupled simulations; e.g., Brun et al. (2017) https://doi.org/10.3189/002214311797409794) and performed point surface energy/mass simulations in the MDVs (line 383).

The authors argue for the standalone approach to "limit uncertainties in meteorological forcing data introduced when coupled to WRF" (line 144), however observational data also contain uncertainties (e.g., the discussion around deriving solid precipitation from SR50s). More importantly, this simplification means that the capability of the full WRF-Hydro/Glacier modelling system has not been assessed and leaves open the question of how the presented modifications impact simulated runoff in coupled simulations when changes in surface conditions can feedback on the atmospheric forcing.

I suggest that the authors either provide a stronger justification for their approach and/or more clearly communicate the novelty and advancement in scientific understanding of their work, or ideally include additional experiments with the full WRF-Hydro/Glacier model (for example, comparing oldrunoff_oldalbedo and newrunoff_newalbedo in an interactive

context). The latter suggestion would greatly strengthen the impact and novelty of the manuscript.

As suggested we have clarified the novelty and advancement of the research presented in our manuscript, as well as explaining the rational of the methodological approach by adding a schematic of the modelling framework and reframing the introduction, defining a new research aim and reiterating how this work has advanced knowledge in the conclusion. Importantly, this research provides a platform to conduct fully distributed hydrological modelling in the MDV.

Research Aim

To ensure readers understand the motivation and importance of this research we have re-written the introduction, and modified the research aim to: "The aim of this study is to optimise a multi-layer snowpack scheme (Crocus), that is integrated in WRF-Hydro/Glacier, to resolve the onset, duration and end of melt over a cold-based glacier in the MDV of Antarctica." It is now clear that the major contribution of this work is to develop the snow and ice modelling component embedded in WRF-Hydro/Glacier, to ensure the physical processes governing melt are resolved in the unique environmental setting of the MDV. What sets this research apart from previous work is that the snow and ice modelling is embedded into WRF-Hydro/Glacier, which allows the routing of meltwater into the surrounding landscape to be resolved for the first time. This research provides the stepping stone to achieve this more ambitious goal.

Novelty

Previous point-scale energy and mass balance modelling in the MDV (Hoffman et al., 2008; Macdonell et al., 2013; Hofsteenge et al., 2022) have used observed albedo to force their models. The use of observed albedo prevents simulations being performed to determine how runoff may change in a future climate and/or extending runoff simulations into areas without albedo measurements. We carefully tune an interactive spectral albedo scheme based on detailed spectral measurements from Antarctica to successfully simulate the temporal evolution of albedo for the first time in the MDV. Importantly, the new modelling framework provides a platform to assess the connections between the atmosphere, glaciers, bare-land surfaces and soils, stream channels and lakes in the future. To do so, we must first modify WRF-Hydro/Glacier for this region. Importantly, our modelling framework allows us to show previously unresolved feedbacks between albedo, melt and snowfall.

As noted by the reviewer, Brun et al. (2011) applied Crocus in Antarctica at Dome C. However, Dome C does not have melt since surface and near surface temperatures never exceed −20 °C. They show that Crocus simulates the snowpack well over a 11-day period, but they do not have melt or wet snow conditions. Here we identify that, in order to successfully apply Crocus to melting conditions on a cold-based Antarctic glacier, it is necessary to modify the percolation of water through ice to correctly represent near surface runoff, in addition to the previously mentioned modifications to the albedo scheme. We show that modelling albedo and near surface runoff are crucial to simulating the evolution of the glacier during the melt season. As noted above, this work provides a platform for future work to improve our understanding of hydrological connectivity in the MDV and other ice-free Antarctic regions.

Methodological choices

We chose to use the WRF-Hydro/Glacier modelling framework rather than just Crocus due to the coupling with the hydrological modules. This modelling framework moves beyond previous work and is capable of pushing the boundaries of any modelling framework for melt and hydrological modelling in this region. This opens up potential for future studies to fill the gap in modelling the hydrological connectivity, however that would not be possible without being able to model the water source on this unique environment. This study shows the necessity of modifying WRF-Hydro/Glacier for cold-based glaciers to accurately simulate melt.

We chose to focus on the snowpack module forced by automatic weather station data at a point on the glacier to limit input uncertainty in the model and demonstrate that the model is able to simulate melt accurately. Using gridded atmospheric data (either from numerical weather prediction models or interpolated from AWS) to force WRF-Hydro/Glacier introduces additional uncertainties. WRF-Hydro is a hydrological model, and it is rarely run in coupled simulations with an atmospheric component (i.e., the WRF model). For example, recent papers focus on so called "standalone simulations" e.g., Xiang et al. (2017), Eidhammer et al. (2021), Lahmers et al. (2021), Pal et al. (2021), Cerbelaud et al. (2022), Mehboob et al. (2022) and many others. We have clarified the choice of model configuration and added a schematic to clarify the modelling framework and which components were used in this study – see the in text changes below. The new schematic clearly shows readers that the primary coupling in WRF-Hydro/Glacier is between the land surface models and terrain routing modules rather than between atmospheric and glacier.

Text changes:

We have provided a stronger justification and highlighted the novelty of this research in the Introduction and Conclusion, including refining our research aim. As noted, we have also added a schematic of the WRF-Hydro/Glacier modelling framework to clarify the model setup and components used in this paper.

We changed the Introduction to:

[revised manuscript text omitted]

**Other key changes**

To ensure the modelling framework is clear to readers we added a new figure of WRF-Hydro/Glacier in Section 2.1. This shows where the coupling is between different modules, and removes any uncertainty about the meteorological forcing data used in this study:

[Figure]

**Figure 1.** Schematic of the WRF-Hydro/Glacier modelling framework. Modules and variables used in this study are displayed in *italics*. Adapted from NCAR (https://ral.ucar.edu/sites/default/files/public/WRFHydroPhysicsComponentsandOutputVariables.png, last access: 8 August 2023).

Changed Line 383: "Although previous studies have implemented surface energy and mass balance models at specific points on MDV glaciers (Hoffman et al., 2008; Macdonell et al., 2013; Hofsteenge et al., 2022), they have neglected to simulate albedo. Implementing WRF-Hydro/Glacier allows us to fill this gap and show that simulating albedo is necessary for simulating the feedbacks between albedo, snowfall and melt in the MDV."

Changed lines 400-408:

**Conclusion**

For the first time, we have simulated the evolution of albedo over a melt season in the MDV. We found it was necessary to modify two schemes to reliably simulate the surface energy balance and runoff of a cold-based MDV glacier:

- The percolation of meltwater through ice layers was modified to allow near-surface runoff, and
- The spectral albedo parameters for both snow and ice were modified based on observed spectral signatures enabling the evolution of broadband albedo and net shortwave radiation to be resolved.

With these modifications, we were able to accurately simulate the feedbacks between albedo, snowfall and melt, which is critical in resolving the onset, duration and end of melt over a cold-based glacier in the MDV. The changes implemented allowed the subtle changes in energy available for melt to be resolved over the course of the ablation season, which would have otherwise not been achievable. Importantly, this is the first time a detailed snowpack model has been coupled to a fully distributed hydrological model in the MDV and represents a significant step towards understanding streamflow dynamics and modelling the full

hydrological connectivity of glacial meltwater in the MDV at different spatial and temporal scales.

Specific comments:

1. The fact that the manuscript presents a standalone simulation without the atmospheric or streamflow components should be more consistently communicated. Although the AWS forcing is mentioned in the abstract, this simplification is not clear from the title or from using the full model name throughout the paper. Statements like „the first application of WRF-Hydro/Glacier model in MDV" (line 5) while technically correct also do not communicate the nuances of the presented work.

As discussed above, WRF-Hydro is primarily a hydrological model, and is not commonly run interactively with the WRF atmospheric model. We have clarified this by adding the schematic of the WRF-Hydro/Glacier modelling framework (Fig. 1) as shown above. We have opted to keep the title as it is important to communicate to readers that we are using the WRF-Hydro/Glacier modelling framework (rather than the snowpack model on its own). We also mention in the title that the model is applied to "a cold-based Antarctic glacier" and that is where the point of interest is located. There are also no on-glacier runoff measurements to validate the runoff from WRF-Hydro/Glacier, so we have opted not to present this.

We have amended Line 5 to: "the first application of WRF-Hydro/Glacier with an embedded multilayer snowpack model at a point in the MDV."

2. Please provide information about the extent and location of the 200-m computational grid and how the point AWS data was distributed spatially over this grid. However, it would be worth mentioning why this grid was used when the two active components (Crocus and Noah-MP) are column models without any lateral interactions and the analysis is point-based.

We agree that mentioning the grid size is confusing to readers and we have removed it as this study is conducted at a point.

We changed Line 145 to: "In this model experiment, we used the Crocus snowpack model embedded in WRF-Hydro/Glacier V5.2.0. Crocus was forced by observed meteorological data at an hourly time step and analyzed at a point on Commonwealth Glacier."

We maintain the description of the distributed model in Section 2.1 as the hydrological modules are spatially distributed and it displays the future uses of the model. The modifications presented in this study can also be applied spatially or in different regions with cold-based glaciers – this is precisely why this model was chosen.

3. Please provide information on time periods for spin-up vs simulation, calibration vs evaluation, and CWG vs COHM forcing earlier in the methods and in one place for convenience (e.g., in a table).

Added to Section 3.3:

**Table 2.** Summary of time periods for model experiment.

|  | Time period | Forcing data |
|---|---|---|
| Spin up | 1 August - 30 November 2021 | COHM AWS |
| Albedo tuning | 1 - 31 December 2021 | CWG AWS |
| Testing | 1 January - 28 February 2022 | CWG AWS |

4. Section 2.1: Please provide more detail about what running in standalone mode means in terms of active components and/or interactions.

As shown above, we have added a schematic of the WRF-Hydro/Glacier modelling framework (Fig. 1) in Section 2.1 to clarify how the components of the model interact and are coupled.

Standalone mode means the land surface model is forced by the atmospheric data, rather than coupled to an atmospheric model. We removed this term in Line 69: "The model can be either forced by meteorological data or other gridded atmospheric data."

To improve the clarity, we also changed Line 144 -145 to: "In this model experiment, we used the Crocus snowpack model embedded in WRF-Hydro/Glacier V5.2.0. Crocus was forced by observed meteorological data at an hourly time step and analyzed at a point on Commonwealth Glacier. The high-quality observational data obtained on Commonwealth Glacier reduces uncertainty that might be introduced by using model or gridded data as forcing data.

5. Line 150: Four months seems short for spinning up ice temperatures in a 50 m column based on my experience with the heat equation. Which objective criteria were used to determine if this period was sufficient?

The purpose of this study is to apply the WRF-Hydro/Glacier modelling system at a point, identify where the model needs modifications for this unique environment and provide robust solutions to ensure the onset, duration and end of melt are resolved. As noted in Lines 149-150, the ice temperatures are initialized at the mean annual temperature of -18 °C, from Obryk et al. (2020).

We changed Line 153 to: We analyzed ice temperatures at the end of the spin-up period and found that the difference between observed and modelled ice temperatures at a depth of 0.05, 0.1, 0.2, 0.5 and 2.0 meters at the beginning of December are less than 1 °C, which is within the sensor uncertainty shown in Table 1."

In Figure 8, it seems the biases change systematically in time (i.e., the cold bias at depth decreases while at upper levels it increases). Could this feature be attributable to spin up (which could be assessed with sensitivity runs) or e.g., the changing depth of the temperature sensors?

TC1 and 2 melted out of the glacier shortly after December 15. Thus, we think that this is likely more due to the changing depth of the sensors, rather than the model spin-up.

6. How was the snow depth and density profile initialized?

Added to Line 148: "Snow depth was initialized as 0 m and the layers were initialized with a constant density of 900 kg m$^{-3}$."

How well are snowpack conditions represented at the end of the spin-up period and how might this influence the simulated albedo?

As mentioned in Q5, the ice temperatures were within the measurement uncertainty of the thermocouples. From the surface height record, we can see that there was a snowfall event at the end of November just before the start of the simulation (also shown in Fig. 9 (now Fig. 10)). At the end of the spin-up period, surface height is 1.4 cm higher than at the start of the spin-up and the top layers are snow with a density of 131 kg/m$^3$, 118 kg/m$^3$ and 786 kg/m$^3$ and the layers below have a density of ice. This ensured that the modelled albedo in Fig. 4 (now Fig. 5) (both in observed and newalbedo) at the start of the simulation is well represented. This is critical for modelling the feedbacks between albedo, snowfall and later in the season, melt.

7. Line 276: how was the absence of overfitting assessed?

We agree that the term overfitting could cause confusion to readers and have removed this. We have removed Line 276 and changed Line 275 to "The newalbedo model better captures the variability in observed albedo over the melt season with a root mean square error of 0.08 compared to oldalbedo with a root mean square error of 0.35."

Suggestions for technical corrections:

1. Line 14: Remove „of melt"

done

2. Table 1: Is it relevant to provide detailed information on instrumentation for COHM as well?

Added to Line 114: "The accuracy of the sensors are similar to CWG AWS and the instruments are detailed in Gooseff et al. (2022)."

3. Line 214: Please clarify that you are referring to the Crocus snow albedo scheme

done

4. Line 300: How do you define slight? The cold bias is ~2 K

Added: "has a cold bias of up to 2 °C"

done

6. Line 328: Rephrase „similar" as surface height changes range approximately an order of magnitude

Changed to: "After 23 December, the rate of decrease in the change in surface height across the three models stabilizes and is less than before. From this point to the end of the simulation, oldrunoff_oldalbedo model has 90 cm ablation, while newrunoff_oldalbedo has 10 cm ablation and newrunoff_newalbedo has 3 cm ablation compared to 10 cm observed ablation."

7. Line 350: Rephrase „ephemeral," as the term may not be accessible to a wide audience

Changed sentence to: "On the other hand, the newrunoff_newalbedo model runoff has less runoff overall and the runoff drops to zero frequently compared to the newrunoff_oldalbedo model, which has runoff every day from the start to end of the season."

---

## Author Comment (AC2)

**Response to reviewer comment #2**. Note reviewers' text is shown in blue, with responses in **black**.

The manuscript submitted by Pletzer et al., is a preliminary step toward applying WRF-Hydro/Glacier to a cold-based Antarctic glacier. Generally, the manuscript is well written, figures and data are presented well and the subject matter is a good fit for the journal. There is potential for this approach to reveal new understanding of the MDV hydrologic system as a whole. However, in agreement with the other reviewer, this present manuscript represents in intermediate methodological step and demonstrates no real advance in scientific understanding as written.

Thank you for taking the time to review this manuscript. This review was helpful in identifying that the aim of the manuscript needed to be refined and that the gap in the literature needed to be more explicitly communicated. This feedback has significantly improved the quality of the manuscript. We have addressed the major comments by rewriting the introduction, research aim and conclusion to explicitly identify the scientific advancement of the manuscript. In addition, we have added a schematic of the WRF-Hydro/Glacier modelling framework to clarify the model setup and components used in this paper. We have also responded to the comments below.

Major Comments:

As this work is presented, the introduction suggests that the full WRF-Hydro/Glacier model is required to make inferences about glacier-stream-lake hydrologic connectivity. However, the rest of the manuscript is focused on the point-based simulation and tuning of Crocus and NoahMP rather than the full model itself. There is no scientific question or hypothesis to be addressed. This paper at a minimum should be reframed to identify and explicitly address the scientific advancement made by this paper alone, not the future applications of the model. A more impactful contribution could involve running experiments with the model.

As suggested, we have clarified the novelty and advancement of the current work, as well as explained the rational of the methodological approach by adding a schematic of the modelling framework and reframing the introduction, research aim, and conclusion. Please refer to the changes suggested in RC1 that address these points.

Minor Comments:

Why was such a short period used for model spin-up? Since the COHM met station was used for spin-up anyway, there are many years of data available from that site. Why limit it to a few months? Also, as the other reviewer notes, might this have some impact on temperature bias shown in the results?

The purpose of this study is to apply the WRF-Hydro/Glacier modelling system at a point, identify where the model needs modifications for this unique environment and provide robust solutions to ensure the onset, duration and end of melt are resolved. As noted in Lines 149-150, the ice temperatures are initialized at the mean annual temperature of -18 °C, from Obryk et al. (2020).

We changed Line 153 to: We analyzed ice temperatures at the end of the spin-up period and found that the difference between observed and modelled ice temperatures at a depth of 0.05, 0.1, 0.2, 0.5 and 2.0 meters at the beginning of December are less than 1 °C, which is within the sensor uncertainty shown in Table 1."

I'd like to see more discussion on how the 2021-22 season relates to the long-term average climate here. Was this a warm, cold, snowy, cloudy etc. season? Since the modified albedo scheme (and overall model tuning and results) were so dependent on data from a single season, it would be nice to provide more context for this season relates to typical summer conditions for this glacier. There is potential to overfit the model for this set of conditions and it may not perform well for colder or warmer seasons. Was this assessed?

We have added the following comparison to show how this season compares to the typical summer to the end of Section 3.3:

"Comparing the 21/22 season to the 1999-2022 long term average over December and January (Hosteenge et al., 2023, *in review*), we find that conditions were typical. Air temperatures were 0.2 °C below average and it was 0.1 m/s less windy. There was less cloud since incoming shortwave radiation was 43.4 W/m$^2$ above average, incoming longwave radiation was 6.9 W/m$^2$ below average and albedo was 0.02 below average."

Please provide more detail on the vertical layer thickness and spacing. Based on figure 3, it is not uniform. This scheme is critical for accurately representing processes that have strong gradients in the shallow subsurface.

We have added this text to Line 87:

"The number of and thickness of vertical layers in Crocus changes dynamically with time. Users define a maximum number of layers (n ≥ 3) and when snowfall occurs, a new layer is added with a set of fresh snow characteristics. Over time, layers may merge with the layer below if the snow grain properties become the same. The layers at the top of the snowpack tend to be thinner to better solve the surface energy balance equation."

How was overfitting assessed for the Albedo modifications?

We agree that the term overfitting could cause confusion to readers and have removed this. We have removed Line 276 and changed Line 275 to "The newalbedo model better captures the variability in observed albedo over the melt season with a root mean square error of 0.08 compared to oldalbedo with a root mean square error of 0.35."

Please provide more detail and comparison of data, instrumentation and accuracy across the two met stations.

Added to Line 112: "The accuracy of the sensors are similar to CWG AWS and the instruments are detailed in Gooseff et al. (2022)."

L121 – Sentence is a repeat from on the previous section

Sentence has been removed.

Section 4.2 – The beginning section as well as a few introductory sentences in paragraphs elsewhere in the section are basically only telling the reader what they will be told later and are therefore unnecessary and should be rewritten.

Removed Lines 200-203 and Line 257.

L355 – this citation is incorrect.

done

---

## Author Response (AR1)

**Response to reviewer comment #1**. Note reviewers' text is shown in blue, with responses in black and changes to the text in green.

General comments:

Pletzer at al. present a point-based optimization and evaluation of a simplified version of the WRF-Hydro/Glacier modelling system over the McMurdo Dry Valleys (MDVs) using forcing data from automatic weather stations. They identify two aspects of the cryospheric component needing improvement for accurate simulations of runoff, namely the representation of percolation in ice layers and the parameters in the albedo scheme. The manuscript is well-written and organized, the results are clearly and concisely presented, and the topic suits the scope of the journal. However, I have a concern about the novelty and wider applicability of the presented results due to the simplified model configuration employed, as outlined in my major comment below, which should be considered prior to publication.

We express our gratitude for taking the time to review our manuscript and offering valuable feedback for its improvement. We have taken your comments into consideration and made significant revisions. Specifically, we have reworked the introduction that includes a new and much clearer research aim, the conclusion to explicitly highlight the scientific advancement of our research and the abstract to communicate the novelty of this study. To eliminate any confusion regarding the motivation for the analysis, we have added a schematic of the WRF-Hydro/Glacier modelling framework. Additionally, we have addressed the minor comments provided. Thank you once again for your time and insightful input.

Major comment:

As I understood, the authors only used the cryospheric (Crocus) and land-surface (Noah-MP) modeling components from WRF-Hydro/Glacier. The presented work is therefore mainly small changes to/calibration of parts of Crocus, which is an important foundational step for tackling interesting science questions in a new region but is itself a methodological task. As a result, the introduction lacks a clear scientific question and, in my opinion, the results may be insufficiently novel, as previous studies have applied Crocus in Antarctica (including in coupled simulations; e.g., Brun et al. (2017) https://doi.org/10.3189/002214311797409794) and performed point surface energy/mass simulations in the MDVs (line 383).

The authors argue for the standalone approach to "limit uncertainties in meteorological forcing data introduced when coupled to WRF" (line 144), however observational data also contain uncertainties (e.g., the discussion around deriving solid precipitation from SR50s). More importantly, this simplification means that the capability of the full WRF-Hydro/Glacier modelling system has not been assessed and leaves open the question of how the presented modifications impact simulated runoff in coupled simulations when changes in surface conditions can feedback on the atmospheric forcing.

I suggest that the authors either provide a stronger justification for their approach and/or more clearly communicate the novelty and advancement in scientific understanding of their work, or ideally include additional experiments with the full WRF-Hydro/Glacier model (for example, comparing oldrunoff_oldalbedo and newrunoff_newalbedo in an interactive

context). The latter suggestion would greatly strengthen the impact and novelty of the manuscript.

As suggested, we have clarified the novelty and advancement of the research presented in our manuscript, as well as explaining the rational of the methodological approach by adding a schematic of the modelling framework and reframing the introduction, defining a new research aim and reiterating how this work has advanced knowledge in the conclusion. Importantly, this research provides a platform to conduct fully distributed hydrological modelling in the MDV.

Research Aim

To ensure readers understand the motivation and importance of this research we have re-written the introduction, and modified the research aim to: "the aim of this study is to optimise a multi-layer snowpack scheme (Crocus), that is embedded in WRF-Hydro/Glacier, to resolve the onset, duration and end of melt over a cold-based glacier in the MDV of Antarctica." It is now clear that the major contribution of this work is to develop the snow and ice modelling component embedded in WRF-Hydro/Glacier, to ensure the physical processes governing melt are resolved in the unique environmental setting of the MDV. What sets this research apart from previous work is that the snow and ice modelling is embedded in WRF-Hydro/Glacier, which allows us to simulate the albedo on an MDV glacier for the first time and resolve the complex feedbacks between albedo, snowfall and melt. This research provides the stepping stone to achieve the more ambitious goal of simulating the routing of meltwater into the surrounding landscape to be resolved.

Novelty

Previous point-scale energy and mass balance modelling in the MDV (Hoffman et al., 2008; Macdonell et al., 2013; Hofsteenge et al., 2022) have used observed albedo to force their models. The use of observed albedo prevents simulations being performed to determine how runoff may change in a future climate and/or extending runoff simulations into areas without albedo measurements. We carefully tune an interactive spectral albedo scheme based on detailed spectral measurements from Antarctica to successfully simulate the temporal evolution of albedo for the first time in the MDV. Importantly, the new modelling framework provides a platform to assess the connections between the atmosphere, glaciers, bare-land surfaces and soils, stream channels and lakes in the future. To do so, we must first modify WRF-Hydro/Glacier for this region. Importantly, our modelling framework allows us to show previously unresolved feedbacks between albedo, melt and snowfall.

As noted by the reviewer, Brun et al. (2011) applied Crocus in Antarctica at Dome C. However, Dome C does not have melt since surface and near-surface temperatures were never warmer than $-20$ °C during the study period. They show that Crocus simulates the snowpack well over a 11-day period, but they do not have melt or wet snow conditions. Here we identify that, in order to successfully apply Crocus to melting conditions on a cold-based Antarctic glacier, it is necessary to modify the percolation of water through ice to correctly represent near-surface runoff, in addition to the previously mentioned modifications to the albedo scheme. We show that modelling albedo and near-surface runoff are crucial to simulating the evolution of the glacier during the melt season. As noted above, this work provides a platform for future work to improve our understanding of hydrological connectivity in the MDV and other ice-free Antarctic regions.

Methodological choices

We chose to use the WRF-Hydro/Glacier modelling framework rather than just Crocus due to the embedding into the hydrological modules. This modelling framework moves beyond previous work and is capable of pushing the boundaries of any modelling framework for melt and hydrological modelling in this region. This opens up potential for future studies to fill the gap in modelling the hydrological connectivity, however that would not be possible without being able to model the water source in this unique environment. This study shows the necessity of modifying WRF-Hydro/Glacier for cold-based glaciers to accurately simulate melt.

We chose to focus on the snowpack module forced by automatic weather station data at a point on the glacier to limit input uncertainty in the model and demonstrate that the model is able to simulate melt accurately. Using gridded atmospheric data (either from numerical weather prediction models or interpolated from AWS) to force WRF-Hydro/Glacier introduces additional uncertainties. WRF-Hydro is a hydrological model, and it is rarely run in coupled simulations with an atmospheric component (i.e., the WRF model). For example, recent papers focus on so called "standalone simulations" e.g., Xiang et al. (2017), Eidhammer et al. (2021), Lahmers et al. (2021), Pal et al. (2021), Cerbelaud et al. (2022), Mehboob et al. (2022) and many others. We have clarified the choice of model configuration and added a schematic to clarify the modelling framework and which components were used in this study – see the in text changes below. The new schematic clearly shows readers that the primary coupling in WRF-Hydro/Glacier is between the land surface models and terrain routing modules rather than between atmospheric and glacier.

Text changes:

We have provided a stronger justification and highlighted the novelty of this research in the introduction and conclusion, including refining our research aim. As noted, we have also added a schematic of the WRF-Hydro/Glacier modelling framework to clarify the model setup and components used in this paper.

We changed the Abstract to:

[revised manuscript text omitted]

**Other key changes**

To ensure the modelling framework is clear to readers we added a new figure of WRF-Hydro/Glacier in Section 2.1. This shows where the coupling is between different modules, and removes any uncertainty about the meteorological forcing data used in this study:

[Figure]

**Figure 1.** Schematic of the WRF-Hydro/Glacier modelling framework. Modules and variables used in this study are displayed in *italics*. Adapted from NCAR (https://ral.ucar.edu/sites/default/files/public/WRFHydroPhysicsComponentsandOutputVaria bles.png, last access: 8 August 2023).

Changed Future research and outlook to:

**6 Future research and outlook**

To better understand the physical processes governing the hydrological cycle in the MDV it is necessary to resolve the connections between the atmosphere, glaciers, bare-land surfaces and soils, stream channels and lakes (hydrological reservoirs). What makes the MDV unique compared to other glaciated environments is that changes in surface energy balance are the primary control on melt water generation and mass balance, as precipitation is limited. Although previous studies have implemented surface energy and mass balance models on glaciers in the MDV glaciers (Hoffman et al., 2008; Macdonell et al., 2013; Hofsteenge et al., 2022) they have not been able to fully account for the meltwater pathways from the glaciers to the surrounding landscape. By implementing a new modelling framework that couples a detailed snowpack model to a fully distributed hydrological model (WRF-Hydro/Glacier), we have taken the first step in enabling the hydrological connectivity of the MDV to be further assessed.

To successfully simulate the onset, duration and end of melt on a cold-based glacier in the MDV, the multi-layer snowpack scheme in WRF-Hydro/Glacier needed to be optimised, which was achieved at a point scale over a 7-month period encapsulating a melt season. The snowpack model (Crocus), which is embedded in WRF-Hydro/Glacier, was (1) modified to limit the percolation of meltwater in the presence of ice layers and (2) optimised to improve the parameter set controlling albedo and net shortwave radiation. We demonstrate that simulating albedo, which has not been attempted before on a glacier in the MDV, is necessary to resolve the complex feedbacks between albedo, snowfall and melt in this energy limited environment. Our approach to simulate albedo based on the evolution of snow grain properties is a significant step forward for modelling glacier response to climate change in the MDV compared to using point-based observations of albedo (e.g. Hoffman (2011); Hofsteenge et al. (2022)).

Future research will be able to utilise this modelling framework to better resolve the spatial and temporal variability in albedo, which is critical in governing spatially distributed melt and hydrological connectivity in the MDV. For example, Bergstrom et al. (2020) measured the spatial variability of albedo from a series of radiometric observations obtained from helicopter flights over three MDV glaciers, which showed that albedo not only increased with elevation but also increased from west to east across both the Canada and Commonwealth glaciers in the Taylor Valley. The longitudinal patterns in albedo observed by Bergstrom et al. (2020) demonstrate that point-based observations of albedo are not sufficient to resolve the spatial and temporal variability in albedo. Thus, the optimised simulation of albedo in the multi-layer snowpack scheme in WRF-Hydro/Glacier used in this study provides a new pathway to resolve this observed complexity, which is critical in governing the amount of meltwater generated from glaciers in the MDV. Importantly, it provides us with confidence that we have developed a modelling framework that will enable us to get the "right answers for the right reasons" (Kirchner, 2006) in regard to resolving one of the key physical processes governing the spatial variability of melt.

The modified WRF-Hydro/Glacier model will also allow us to expand on the work from Hoffman (2011) and Cross et al. (2022) by accounting for in-stream processes such as evaporation and soil absorption of meltwater. These processes are expected to have the greatest impact on streamflow in low melt years and in the larger, more complex tributaries.

Furthermore, explicitly modelling the stream channels will allow us to answer questions about the timing between melt generation and lake inflow, which has downstream impacts on nutrient availability for the microbial ecosystems found in the MDV (Gooseff et al., 2017; Singley et al., 2021). By having the ability to better understand streamflow dynamics and hydrological connectivity, it is anticipated future studies using this modelling framework will be capable of providing new insights into the impacts of climate forcing on meltwater generation.

Specific comments:

1. The fact that the manuscript presents a standalone simulation without the atmospheric or streamflow components should be more consistently communicated. Although the AWS forcing is mentioned in the abstract, this simplification is not clear from the title or from using the full model name throughout the paper. Statements like „the first application of WRF-Hydro/Glacier model in MDV" (line 5) while technically correct also do not communicate the nuances of the presented work.

As discussed above, WRF-Hydro is primarily a hydrological model, and is not commonly run interactively with the WRF atmospheric model. We have clarified this by adding the schematic of the WRF-Hydro/Glacier modelling framework (Fig. 1) as shown above. We have opted to keep the title as it is important to communicate to readers that we are using the WRF-Hydro/Glacier modelling framework (rather than the snowpack model on its own). We also mention in the title that the model is applied to "a cold-based Antarctic glacier" and that is where the point of interest is located. There are also no on-glacier runoff measurements to validate the runoff from WRF-Hydro/Glacier, so we have opted not to present this.

We have amended Line 5 (now 6) to: "To establish a new framework to do this, we present the first application of WRF-Hydro/Glacier in the MDV, which as a fully distributed hydrological model, has the capability to resolve the pathways of meltwater from the glaciers to the bare-land surfaces that surround them."

2. Please provide information about the extent and location of the 200-m computational grid and how the point AWS data was distributed spatially over this grid. However, it would be worth mentioning why this grid was used when the two active components (Crocus and Noah-MP) are column models without any lateral interactions and the analysis is point-based.

We agree that mentioning the grid size is confusing to readers and we have removed it as this study is conducted at a point.

We changed Line 145 (now 194-195) to: "In this model experiment, we used the Crocus snowpack model embedded in WRF-Hydro/Glacier version 5.2.0 (Eidhammer et al., 2021). Crocus was forced by observed meteorological data at an hourly time step and analyzed at a point on Commonwealth Glacier."

We maintain the description of the distributed model in Section 2.1 as the hydrological modules are spatially distributed and it displays the future uses of the model. The modifications presented in this study can also be applied spatially or in different regions with cold-based glaciers – this is precisely why this model was chosen.

3. Please provide information on time periods for spin-up vs simulation, calibration vs evaluation, and CWG vs COHM forcing earlier in the methods and in one place for convenience (e.g., in a table).

Added to Section 3.3:

**Table 2.** Summary of time periods for model experiment.

|  | Time period | Forcing data |
|---|---|---|
| Spin up | 1 August - 30 November 2021 | COHM AWS |
| Albedo tuning | 1 - 31 December 2021 | CWG AWS |
| Testing | 1 January - 28 February 2022 | CWG AWS |

4. Section 2.1: Please provide more detail about what running in standalone mode means in terms of active components and/or interactions.

As shown above, we have added a schematic of the WRF-Hydro/Glacier modelling framework (Fig. 1) in Section 2.1 to clarify how the components of the model interact and are coupled.

Standalone mode means the land surface model is forced by the atmospheric data, rather than coupled to an atmospheric model. We removed this term in Line 69 as it can be confusing to readers.

To improve the clarity, we also changed Line 144 -145 (now 195-197) to: "Crocus was forced by observed meteorological data at an hourly time step and analyzed at a point on Commonwealth Glacier. The high-quality observational data obtained on Commonwealth Glacier reduces uncertainty that might be introduced by using model or gridded data as meteorological forcing data."

5. Line 150: Four months seems short for spinning up ice temperatures in a 50 m column based on my experience with the heat equation. Which objective criteria were used to determine if this period was sufficient?

The purpose of this study is to apply the WRF-Hydro/Glacier modelling system at a point, identify where the model needs modifications for this unique environment and provide robust solutions to ensure the onset, duration and end of melt are resolved. As noted in Lines 149-150 (now 200), the ice temperatures are initialized at the mean annual temperature (Fountain et al, 1998) of -18 °C.

We changed Line 153 (now 205-208) to: "We analyzed ice temperatures at the end of the spin-up period and found that the difference between observed and modelled ice temperatures at a depth of 0.05, 0.1, 0.2, 0.5 and 2.0 meters at the beginning of December were less than 1 °C, which is within the sensor uncertainty shown in Table 1. Given this agreement, we concluded the spin-up time is sufficient for the model testing in this evaluation (see Section 5.2 for further discussion)."

In Figure 8, it seems the biases change systematically in time (i.e., the cold bias at depth decreases while at upper levels it increases). Could this feature be attributable to spin up (which could be assessed with sensitivity runs) or e.g., the changing depth of the temperature sensors?

TC1 and 2 melted out of the glacier shortly after December 15. Thus, we think that this is likely more due to the changing depth of the sensors, rather than the model spin-up.

6. How was the snow depth and density profile initialized?

Added to Line 148 (now 199): "Snow depth was initialized as 0 m and the layers were initialized with a constant density of 900 kg m$^{-3}$."

How well are snowpack conditions represented at the end of the spin-up period and how might this influence the simulated albedo?

As mentioned in Q5, the ice temperatures were within the measurement uncertainty of the thermocouples. From the surface height record, we can see that there was a snowfall event at the end of November just before the start of the simulation (also shown in Fig. 9 (now Fig. 10)). At the end of the spin-up period, surface height is 1.4 cm higher than at the start of the spin-up and the top layers are snow with a density of 131 kg/m$^3$, 118 kg/m$^3$ and 786 kg/m$^3$ and the layers below have a density of ice. This ensured that the modelled albedo in Fig. 4 (now Fig. 5) (both in observed and newalbedo) at the start of the simulation is well represented. This is critical for modelling the feedbacks between albedo, snowfall and slightly later in the season, melt.

7. Line 276: how was the absence of overfitting assessed?

We agree that the term overfitting could cause confusion to readers and have removed this. We have removed Line 276 and changed Line 275 (now 325-327) to "Variability in observed albedo is better captured by newalbedo over the melt season with a root mean square error of 0.08 compared to oldalbedo with a root mean square error of 0.35. Accurately simulating albedo enables us to better simulate the feedbacks between albedo, precipitation and melt."

Suggestions for technical corrections:

1. Line 14: Remove „of melt"

Corrected

2. Table 1: Is it relevant to provide detailed information on instrumentation for COHM as well?

Added to Line 114 (now 153-154): "The accuracy of the sensors are similar to CWG AWS and the instruments are detailed in Gooseff et al. (2022)."

3. Line 214: Please clarify that you are referring to the Crocus snow albedo scheme

Corrected

4. Line 300: How do you define slight? The cold bias is ~2 K
Added to Line 300 (now 351): "has a cold bias of up to 2 °C"

5. Line 307: „a cold bias and a phase shift"

Corrected

6. Line 328: Rephrase „similar" as surface height changes range approximately an order of magnitude

Changed Line 328 (now 378-380) to: "After 23 December, the rate of decrease in the change in surface height across the three models stabilizes. From this point to the end of the simulation, oldrunoff_oldalbedo has 90 cm ablation, while newrunoff_oldalbedo has 10 cm ablation and newrunoff_newalbedo has 3 cm ablation compared to 10 cm observed ablation."

7. Line 350: Rephrase „ephemeral," as the term may not be accessible to a wide audience

Changed Line 350 (now 401-403) to: "There is less runoff overall in newrunoff_newalbedo and runoff drops to zero frequently compared to newrunoff_oldalbedo, which has runoff every day from the start to the end of the season."

**Response to reviewer comment #2**. Note reviewers' text is shown in blue, with responses in black and changes to the text in green.

The manuscript submitted by Pletzer et al., is a preliminary step toward applying WRF-Hydro/Glacier to a cold-based Antarctic glacier. Generally, the manuscript is well written, figures and data are presented well and the subject matter is a good fit for the journal. There is potential for this approach to reveal new understanding of the MDV hydrologic system as a whole. However, in agreement with the other reviewer, this present manuscript represents in intermediate methodological step and demonstrates no real advance in scientific understanding as written.

Thank you for taking the time to review this manuscript. This review was helpful in identifying that the aim of the manuscript needed to be refined and that the gap in the literature needed to be more explicitly communicated. This feedback has significantly improved the quality of the manuscript. We have addressed the major comments by rewriting the introduction, research aim and conclusion to explicitly identify the scientific advancement of the manuscript. In addition, we have added a schematic of the WRF-Hydro/Glacier modelling framework to clarify the model setup and components used in this paper. We have also responded to the comments below.

Major Comments:

As this work is presented, the introduction suggests that the full WRF-Hydro/Glacier model is required to make inferences about glacier-stream-lake hydrologic connectivity. However, the rest of the manuscript is focused on the point-based simulation and tuning of Crocus and NoahMP rather than the full model itself. There is no scientific question or hypothesis to be addressed. This paper at a minimum should be reframed to identify and explicitly address the scientific advancement made by this paper alone, not the future applications of the model. A more impactful contribution could involve running experiments with the model.

As suggested, we have clarified the novelty and advancement of the current work, as well as explained the rational of the methodological approach by adding a schematic of the modelling framework and reframing the introduction, research aim, and conclusion. Please refer to the response and changes suggested to the major comment of Reviewer #1 that address these points.

Minor Comments:

Why was such a short period used for model spin-up? Since the COHM met station was used for spin-up anyway, there are many years of data available from that site. Why limit it to a few months? Also, as the other reviewer notes, might this have some impact on temperature bias shown in the results?

The purpose of this study is to apply the WRF-Hydro/Glacier modelling system at a point, identify where the model needs modifications for this unique environment and provide robust solutions to ensure the onset, duration and end of melt are resolved. As noted in Lines 149-150 (now 200), the ice temperatures are initialized at the mean annual temperature (Fountain et al, 1998) of -18 °C.

We changed Line 153 (now 205-208) to: "We analyzed ice temperatures at the end of the spin-up period and found that the difference between observed and modelled ice temperatures at a depth of 0.05, 0.1, 0.2, 0.5 and 2.0 meters at the beginning of December were less than 1 °C, which is within the sensor uncertainty shown in Table 1. Given this agreement, we concluded the spin-up time is sufficient for the model testing in this evaluation (see Section 5.2 for further discussion)."

I'd like to see more discussion on how the 2021-22 season relates to the long-term average climate here. Was this a warm, cold, snowy, cloudy etc. season? Since the modified albedo scheme (and overall model tuning and results) were so dependent on data from a single season, it would be nice to provide more context for this season relates to typical summer conditions for this glacier. There is potential to overfit the model for this set of conditions and it may not perform well for colder or warmer seasons. Was this assessed?

We have added the following comparison to show how this season compares to the typical summer to the end of Section 3.3:

"Comparing the 21/22 season to the 1999-2022 long term average over December and January (Hofsteenge et al., 2023, *in review*), we find that air temperature, wind and albedo were close to average. Air temperatures were 0.2 °C below average, windspeed was 0.1 m s$^{-1}$ below average and albedo was 0.02 below average. There was slightly less cloud cover as incoming shortwave radiation was 43.4 W m$^{-2}$ (13.9%) above average and incoming longwave radiation was 6.9 W m$^{-2}$ (-3.0%) below average."

Please provide more detail on the vertical layer thickness and spacing. Based on figure 3, it is not uniform. This scheme is critical for accurately representing processes that have strong gradients in the shallow subsurface.

We have added this text to Line 87 (now 125-128):

"The number of and thickness of vertical layers in Crocus changes dynamically with time. Users define a maximum number of layers (n ≥ 3) and when snowfall occurs, a new layer is added with a set of fresh snow characteristics. Over time, layers may merge with the layer below if the snow grain properties become the same. The layers at the top of the snowpack tend to be thinner to better solve the surface energy balance equation."

How was overfitting assessed for the Albedo modifications?

We agree that the term overfitting could cause confusion to readers and have removed this. We have removed Line 276 and changed Line 275 (now 325-327) to "Variability in observed albedo is better captured by newalbedo over the melt season with a root mean square error of 0.08 compared to oldalbedo with a root mean square error of 0.35. Accurately simulating albedo enables us to better simulate the feedbacks between albedo, precipitation and melt."

Please provide more detail and comparison of data, instrumentation and accuracy across the two met stations.

Added to Line 114 (now 153-154): "The accuracy of the sensors are similar to CWG AWS and the instruments are detailed in Gooseff et al. (2022)."

L121 – Sentence is a repeat from on the previous section

 Sentence has been removed.

Section 4.2 – The beginning section as well as a few introductory sentences in paragraphs elsewhere in the section are basically only telling the reader what they will be told later and are therefore unnecessary and should be rewritten.

Removed Lines 200-203 and Line 257.

L355 – this citation is incorrect.

Corrected

---

## Author Response (AR2)

**Response to reviewer comment #2.** Note reviewers' text is shown in blue, with responses in **black**.

Several times the authors mention "meltwater pathways" from the glaciers to the landscape. I think they should add more clarification on that phrase. I.e. physical routing of water? The model grid cell resolution certainly cannot handle the supraglacial drainage network on these glaciers (I'm curious to see how they will handle that in the future). Or do they mean the ways in which meltwater is generated? i.e. ice surface or subsurface?

We agree this can cause confusion and have changed "meltwater pathways" to "stream channels and streamflow" in Line 62 and 436. We have also changed "pathways of meltwater" in Lines: 7, 56 and 83 to "streams".

This modification to the model was optimized for a very specific point location on the Commonwealth Glacier, a relatively cold, low-melt, location. While the overarching processes and modifications will hold, there is a lot of variability across glaciers in the MDV. I think the discussion should include some acknowledgement and discussion of how different the energy balance dynamics might be across glaciers in the MDVs and how this will be taken into consideration in the expansion of this model.

This is an interesting consideration, however it is beyond the scope of this paper.

Several times, the authors mention "bare land surfaces and soils". I'm confused what they mean by this since all locations without ice, snow or lake, are bare land. Most of those locations have soils except for some exposed bedrock. What is the point of this phrasing?

We have changed "bare land surfaces and soils" to "bare land" in Lines 7, 57, 60, 138, 140 and 432.

The "bare land" specifically refers to the USGS 24-type land use land cover product used in the land surface model. WRF-Hydro/Glacier can be run over different land surface types (including vegetation).

There are multiple locations where the authors have sentences or whole paragraphs laying out the content of the paper or section. I think this is unnecessary as each section header effectively does that job. These locations include: the last paragraph of the introduction, the introductory paragraph to section 4, the first paragraph of section 4.2, and the introductory paragraph to section 5.

We kept the last paragraph of the introduction because it outlines the structure of the paper. As suggested, we have removed the other paragraphs mentioned.

Note to the Editor: if you feel strongly that the last paragraph in the introduction should be removed, then I am also comfortable with it being deleted.

I recommend designating what was used in this model in italics and bold in figure 1. It is not obvious enough with italics alone.

We have underlined the italics in Figure 1 to highlight what components of WRF-Hydro/Glacier were used. Bold was confusing with the different model names which are also in bold.

In section 6 the authors use the term "soil absorption of meltwater". I believe what they are actually referring to is transmission losses of water to the hyporheic zone in stream channels. There are several papers that describe this process well for the dry valley streams. I recommend the authors cite them and use proper terminology.

WRF-Hydro/Glacier does not model the hyporheic zone at this point, though this can be an important process in longer streams. The "soil absorption of meltwater" means water that is absorbed by the soil.